# Wastepaper-Based Cuprammonium Rayon Regenerated Using Novel Gaseous–Ammoniation Injection Process

**DOI:** 10.3390/polym16172431

**Published:** 2024-08-27

**Authors:** Sherif S. Hindi

**Affiliations:** 1Department of Agriculture, Faculty of Environmental Sciences, King Abdulaziz University (KAU), P.O. Box 80208, Jeddah 21589, Saudi Arabia; shindi@kau.edu.sa; Tel.: +966-566-760-086; 2Center of Excellence in Environmental Studies, KAU, P.O. Box 80216, Jeddah 21589, Saudi Arabia

**Keywords:** wastepaper, cuoxam solution, gaseous ammoniation process, cuprammonium rayon, hardening solution, fabrics

## Abstract

Rayon is an extremely valuable cellulosic fiber in the global textile industry. Since cuprammonium rayon is more eco-friendly than other types of rayon fabrics, it was synthesized by regenerating α–cellulose isolated from wastepaper using a novel gaseous-ammoniation injection (GAI) process. This was achieved by preparing tetra–ammine copper hydroxide (cuoxam solution) via reacting copper sulfate and sodium hydroxide to produce copper hydroxide that was, finally, ammoniated by injecting the gas directly to the reaction vessel instead of using ammonium hydroxide applied by prior art. After that, the air-dried cellulose was chemically generated by dissolving it in a freshly prepared cuoxam solution and, subsequently, was regenerated by extruding it within a hardening bath constituted mainly from citric acid, producing the cuprammonium rayon (c. rayon). The properties of the fibrous, structural (XRD and mechanical), physical, and chemical features were investigated. It was found that the rayon was produced in a high yield (90.3%) with accepted properties. The fibrous properties of the rayon staple length, linear density, and fiber diameter were found to be 44 mm, 235 Tex, and 19.4 µm, respectively. In addition, the mechanical properties determined, namely tensile strength, elongation at break, modulus of elasticity, and breaking tenacity, were found to be 218.3 MPa, 14.3 GPa, 16.1%, and 27.53 cN/Tex, respectively. Based on this finding, and upon injecting the ammonia gas through the α–cellulose saturated and immersed in the Cu (OH)_2_ to complete producing the cuoxam solvent, we find that theuse of an injection rate of 120 mL/minute to obtain the highest fibers’ tensile strength for the final product of the c. rayon is preferable. Utilization of higher rates will consume more amounts of the ammonia gas without gaining noticeable enhancement in the c. rayon’s mechanical quality. Accordingly, the GAI invention rendered the c. rayon favorable for use in making sustainable semisynthetic floss for either insulation purposes or spun threads for woven and nonwoven textile clothing.

## 1. Introduction

Using renewable and sustainable resources in industry has been an attractive target due to their biodegradability, their being nonpetroleum-based, carbon neutral, and having low environmental risks and controlled health hazards [1].

Cellulose is the most abundant natural macromolecule found in Nature. Wood is the main precursor of cellulose, which contains about 40% to 50% content, along with lignin, hemicelluloses, organic extractives, and minerals [2,3]. The annual production of cellulose was estimated to be about 7.5 × 10^10^ t [4]. Cellulose and its derivatives are of great importance in various areas of human activities. Native cellulose fibers are used essentially in textile, pulp, and paper industries and as composite materials [4,5,6,7].

Wood waste from annual pruning and lignocellulosic recycled materials is gaining sustainable interest due to the scarcity of traditional fibrous materials and increasing demand, especially in Saudi Arabia.

While there is no single, universally accepted concentration limit for calcium carbonate in rayon, generally accepted limits range from 0.1% to 1.0% by weight of the fiber, depending on the application, quality requirements, and regulatory standards. 

Cellulose solubility was found to be influenced by several factors, such as fiber morphology, molecular weight, degree of organization, origin of the polymer, and hydrogen bonding network, which affect the quality and performance of cellulosic products [5,8,9,10,11].

Finding an efficient solvent system for cellulose is a long-standing goal in cellulose research and industry. Due to the complexity of the cellulose network, the partial crystalline structure, and the extended noncovalent interactions among molecules (hydrogen bonding and van der Waals forces), the chemical processing of cellulose makes this task more difficult. Cellulose is neither meltable nor soluble in water or in a large range of organic compounds [12]. Cellulose solutions are usually divided into two main groups: nonderivatizing (via physical intermolecular interactions) and derivatizing solutions via covalent modification [13]. The viscose process using NaOH and carbon bisulfide (CS_2_) is the most common derivatizing cellulose solution system used [14]. The nonderivatizing solution class for cellulose comprises systems capable of dissolving cellulose only via physical intermolecular interactions. Cuprammonium hydroxide, simply termed as cuoxam (Schweitzer’s) solution, was reported to be among the most popular solvents of cellulose [15,16]. Later, ethylenediamine was found to be a good alternative to ammonia [17,18,19]. Several alternative systems have been reported, which use transition metals such as palladium and zinc as well as amine or ammonium compounds.

Although chemically comparable to cotton, rayon is a regenerated cellulose fiber that is different from cotton in that its molecular weight is approximately one-fifth and its crystallinity is approximately one-half of cotton’s [20]. Rayon is derived from the French word “rays of light” and was first sold as artificial silk [21]. Fabrication of rayon through the xanthate includes using toxic carbon disulfide. The cuprammonium process is one of the earliest methods for producing rayon and is more eco-friendly as well as more cost-effective than the other methods [16]. Besides their use in the textile industry, rayon textile-reinforced composites are essential for tires, conveyor belts, hoses, and V-belts [22,23,24,25,26,27,28,29,30].

Rayon fibers can be synthesized by one of three methods leading to three distinct and different types, namely the viscose rayon, the cuprammonium rayon, and the saponified cellulose acetate [31]. The viscose process is reasonably cost-effective and holds particular importance in the manufacturing of nonwoven fabrics [31,32]. The cuprammonium technique is a valuable method for producing rayon from textile waste. The process of producing rayon from various waste materials entails the conversion of cellulose into a liquid compound, followed by its reversion back into cellulose in the form of a fiber. Cellulose undergoes a reaction with liquid ammonia and basic cupric carbonate, resulting in the formation of a soluble polymer that can be further transformed into rayon [16]. Rayon is classified as a regenerated fiber since it undergoes a process where cellulose is transformed into a liquid component and subsequently reconverted back into cellulose in the form of fiber.

At present, the cuprammonium rayon, also termed as cupro fabric, is made by dissolving cotton linter or wood pulp in an ammoniacal copper sulfate solution [16,31,33].

The rayon fiber has the ability to be spun into a very thin denier and exhibits softness and handling properties similar to silk fiber. This fiber is utilized in comparable applications to cellulose acetate fibers [7].

According to the research conducted by Sobue et al. [34], it has been established that cellulose can be dissolved in water-based sodium hydroxide (NaOH) at temperatures below –5 °C as long as the NaOH concentration is within the range of 7–10 [34,35]. The aqueous alkali systems frequently fail to fully disintegrate the semicrystalline portions of cellulose, and their ability to dissolve cellulose is restricted to types with a relatively low degree of polymerization (DP). The apparent solubility of cellulose is influenced by both the degree of crystallinity and the kind of crystal. The successful utilization of pretreatments, such as steam explosion, on dissolving pulp has been documented by Yamashiki et al. [36]. Additional water-based alkaline substances, such as LiOH [37] or quaternary ammonium hydroxides, also have the ability to dissolve cellulose [4,17,38]. In recent times, researchers have investigated the use of aqueous NaOH solutions in combination with additional substances such as polyethylene glycol [39], urea [40], and thiourea [15,41]. Cuoxam [16] was the first to report the successful dissolution of cellulose. It was reported that a solution of cupric hydroxide in aqueous ammonia has the ability to dissolve cellulose. The commonly used term for this solution is the cuoxam procedure. The cuprammonium procedure chemically transforms cellulose into a soluble substance by reacting it with copper and ammonia [16]. Rayon is produced by dissolving cellulose in a solution called cuoxam, which is a deep blue solution (turquoise color) containing tetra–ammine cupric hydroxide, also known as cuoxam (Schweitzer) reagent. The latter is achieved by combining a copper sulfate solution with NaOH to form cupric hydroxide, which is then dissolved in NH_4_OH solution and extruded through a spinneret into a hardening bath known as coagulation bath hardening process typically involves immersing the fibers in a chemical solution that contains a coagulant such as sulfuric acid, aluminum sulfate, or others. The coagulant reacts with the fiber’s surface, causing the molecules to aggregate and form a more rigid structure. The exact conditions of temperature, concentration, and duration of the treatment can be tailored to achieve specific properties for different results.

This process is necessary to break down the cuprammonium complex and yield cellulose [31]. This method is costlier than the one used for viscose rayon. The fiber has a nearly round cross-section [16,42]. The basic advantage of using these materials is the promotion of eco-friendly materials.

For the preparation of the cuoxam solution, Sharma et al. [31] dissolved about 20 g of copper sulfate in 100 mL distilled water and 15 mL of dilute H_2_SO_4_ while stirring the solution clearance. After that, the resultant precipitate of cupric hydroxide was dissolved in 11 mL of liquor ammonia until a deep blue solution (turquoise color) of the cuoxam solution was obtained.

A yarn is a long and slender product made of fibers and/or filaments, with or without twist, that is used for interlacing in activities like weaving, knitting, or sewing [43]. Staples are fibers with discrete lengths regardless of their composition, while filaments have continuous to near-continuous lengths [44].

Linear density, yarn count, yarn number, and yarn size are all indicators of the thinness or thickness of a yarn. The measurement of yarn fineness cannot be accurately determined by its diameter due to the inconsistent and nonuniform nature of its diameter along its length, as well as the possibility of its cross-sectional shape not being circular [45]. The linear density of fibers refers to the weight of fibers in grams per unit length, which can vary depending on the measuring system employed. There are three basic methods for this measurement, namely, Tex, deciTex, and denier systems. The Tex system is defined as the mass in grams per 1000 m, denier is the mass in grams per 9000 m, while deciTex (dTex) denotes the mass in grams per 10,000 m.

Since finding an efficient solvent system of cellulose is a long-standing goal in cellulose research and industry, the cuprammonium rayon was synthesized, and its ability to dissolve recycling wastepaper in an ammoniacal copper sulfate solution [16,31,33] as well as its suitability to be spun into a very thin denier exhibiting softness and handling properties.

## 2. Materials and Methods

### 2.1. The Management Plan

As seen in Figure 1, the management strategy is depicted, using the novel synthesis of cuprammonium rayon from recycled wastepaper precursor.

### 2.2. Raw Materials

At first, wastepaper was collected, sorted, cleaned and subsequently was subjected to purification process to extract pure alpha cellulose.

#### Eliminating Calcium Carbonate from the Crude Wastepaper

Calcium carbonate (CaCO_3_) was dissolved at various concentrations of HCl (0.50, 1.00, and 2.50 wt%) under different pressures. Under reduced pressure, the total amount of the acid consumed was reduced while maintaining a high dissolving rate [46]. They explained that the increase in the dissolving rate of the CaCO_3_ with decreasing pressure was due to le Chatelier’s Law, the stirring effect due to rising bubbles, and the cavitation effect. As indicated by Takahashi et al. [46], the chemical reaction between calcium carbonate and hydrochloric acid took place according to the following equation:CaCO_3_ + 2H^+^→ Ca^2+^ + H_2_ O + CO_2_↑.

It can be indicated from Figure 2 that the crude macerated cellulosic fibers (Figure 2a) still had calcium carbonate particles, while those presented at Figure 2b was in a pure state due to the chemical treatment performed.

### 2.3. Chemical Reagents (CR) For the C. Rayon Synthesis

Seven chemical reagents were used in this study to synthesize the rayon fibers (Table 1). Four of them were purchased in a commercial grade, namely copper sulfate, sodium hydroxide, ammonia gas, and citric acid. The reminder CR, namely copper hydroxide and cuoxam solution, were laboratorially prepared.

### 2.4. Synthesis of the C. Rayon Fibers

Concerning the synthesis of the cellulose’s solvent (cuoxam reagent), about 50 g of copper sulfate pentahydrate (CuSO_4_) was dissolved in one liter of deionized water to obtain 5% concentration. It was formed due to the reaction between copper sulfate and caustic soda at room temperature with continuous stirring [7]. In the present study, about 5% wt/wt of citric acid was added to the CuSO_4_ solution to prevent its hydrolysis.

In addition, about 10 g of the NaOH was well-dissolved in one liter of deionized water to obtain a 1% concentration and was mixed with the CuSO_4_ solution in a ratio of 1:1 (Figure 3d–h). After the complete dissolving, the supernatant was filtered using a polypropylene textile with an average pore diameter of about 10–20 µm and stored until used. The cupric hydroxide precipitate was washed sufficiently with deionized water until the filtrate fails to give a positive test for sulfate ions with barium chloride solution.

Regarding the novel process termed as gaseous ammoniation of the copper hydroxide, ammonia gas was purchased from Abdullah Hashim Industrial Gases & Equipment Co., Ltd., Jeddah, Saudi Arabia, so this reaction must be performed in good ventilation conditions and/or in a fume hood to limit exposure to its distinct odor (Figure 3c–e).

This step upon synthesizing the cellulose solvent (cuoxam solution) is the cornerstone of the present investigation (Figure 3 and Appendix A). In all previous prior art, ammonium hydroxide solution has been utilized for ammoniation of the copper hydroxide in order to graft four NH_3_ groups on the Cu(OH)_2_ molecule. In the meantime, this study has grafted the four ammonia groups via adding the ammonia gas directly to the Cu(OH)_2_ paste, as shown in Figure 3c–e. In addition, the gas can be pressurized into the Cu(OH)_2_ paste either within a pressure vessel (Figure 3a) to accelerate the ammoniation process rate as well as save the gas lost or in an open system like a baker (Figure 3b).

It is worth mentioning that the synthesis of the cuoxam solvent must be performed under cooling since this reaction is exothermic. It is featured by its dark blue aspect (turquoise color), as is clearly seen in Figure 3f. The process can be performed in a traditional vessel or in a special one modified to be safer for such exothermic reactions [47]. The neutralized pasty precipitate was treated by ammonia gas until turning its turquoise color into a deep blue that featured the solution of tetra–ammine cupric hydroxide (cuoxam solution).

In this wet spinning, a cuoxam solution containing the dissolved cellulose was controlled–deposited by using a syringe pump and extruded through its fine nozzle. The fibers were regenerated soon after immersing in the coagulation bath, termed as hardening or curing bath, where the cellulose polymer precipitates (Figure 3h). The hardening bath was prepared to be citric acid (10% wt/wt). The main advantage of this method is that there is no thermal degradation of the cellulosic material, and a smaller fiber diameter can be obtained [48].

A known weight of α–cellulose was dissolved in the cuoxam solution with continues stirring using a magnetic stirrer. This solution was called the uncured rayon solution. The solution was filtered using a standard sieve (80 mesh) to eliminate undissolved cellulosic particles. After the filtration, the viscosity of the resultant pasty solution was adjusted since this viscosity affects the thickness of the resultant threads. If thick threads are wanted, high viscous solution must be prepared and vice versa.

In this investigation, the hardening bath for the rayon was citric acid (C_6_H_8_O_7_) solution (5%, wt/wt) since it is safer than the ordinary bath of sulphuric acid (10%, wt/wt) and is needed for public health considerations. Such a hardening bath can be reused multiple times with frequent adjustment of its pH. It is worth mentioning that the hardening bath must be cooled by ice or liquid nitrogen since the curing reaction is exothermic or spontaneous; this results in an excess release of heat energy, making the fibers weak and breakable. In addition, the hardening container had an inner perforated vessel. This facilitated the exit of the cured rayon fibers from the curing solution and the quick transferring to the subsequent washing step.

To press the rayon fibers, the spinning dope is released through the spinneret holes into the coagulation bath, resulting in the creation of rather thick filaments. These filaments are then stretched to decrease their thickness. Figure 3 displays the reinforced rayon fibers [7]. For this purpose, automatic spinnerets were filled with well-filtered uncured rayon solution.

The rayon fibers were allowed to stand in the hardening solution for about 15 min until a decolorization from blue to opaque white occurred and until they became strong enough. As shown in Figure 3h, the recently pressed threads within the citric acid bath appeared in a blue color, while they converted to a white color when they lost their copper ions.

Mimicking some old recipes of the prior art that suggested adding a definite amount of oxalic acid and glyceric acid, we added about glucose (5%, wt/wt) to the citric acid bath in a ratio of 1:10 in order to make the dope solution more spinnable and provide the filaments with superior uniformity.

### 2.5. Fabrication of the C. Rayon Products

#### 2.5.1. Production of Woven Fabric

The elementary products of rayon in this invention were short fiber–staple, long fiber–staple, and filament fibers, while the final products were nonwoven and woven textiles. However, to achieve these products, carding, spinning, and weaving processes were applied using primitive tools.

##### The Carding Process

The first step in this industry is the carding process, in which the random rayon fibers are ordered and aligned by using a primitive carding machine [49], as shown in Figure 4.

##### The Spinning Process

It is the consequent process after the carding one, whereby staple fibers were converted into threads by using manual twisting beside a manual hand spindle (Figure 4). The twisting process was performed to enhance the strength of the inter-fiber cohesion. The rayon yarn’s strength and flexibility are extensively reviewed to be dependent on several factors, namely the degree of fiber-to-fiber overlap, the surface characteristics of the fibers, the degree of twist, the tightness of the twist, and the fiber strength [49].

##### The Weaving Process

Weaving is known as the fabric manufacturing method, and it is applied in the present study using a primitive weaving machine (Figure 4). The fabric was made from two unique sets of threads, which were intertwined at right angles to construct a woven fabric.

During the weaving process, weft yarns are interlaced between two layers of warp yarns at a right angle of 90° to the warp strands. Two weaving structures (stretch patterns) were constructed in this study (Figure 4), namely, plain weave 1/1 and Panama weave 2/2. Accordingly, two different types of textile architecture have been fabricated: (a) Plain weave 1/1 in which the yarn was crossing over one warp yarn; (b) Panama weave 2/2 in which the weft yarns are not crossing every warp yarn [50]. Both architectures were different in their yarn density (the number of yarns regarding their width).

#### 2.5.2. Production of Nonwoven Fabric

Nonwoven fabrics consist of randomly arranged sheets of fibers that are bonded together through adhesive bonding, entanglement, or sewing. Cords and ropes utilize two-dimensional (2D) constructions. Three-dimensional (3D) fabrics are utilized as composite preforms, either in the configuration of molded sheets or substantial constructions. Additionally, they find application in other areas such as knitted clothing and conveyor belts [51]. The nonwoven textile was fabricated in this investigation by extrusion of the rayon fibers within the curing or hardening solution on a perforated plate in a duplicate layer whereby any sublayer was perpendicular to the other one (Figure 4). No adhesives were used to bind the fibers into the 3D structure of the nonwoven fabric. Welding the loosened permeable spot shown in Figure 4(b1) into a sheet (Figure 4(b2)) was achieved using a thermal spot-welding machine using red copper protrusions. Accordingly, loosened permeable spots (Figure 4(b4–b7)) and welded spots (Figure 4(b5,b6)) were obtained, thus offering good permeability and reinforcing the nonwoven fabric, respectively.

It is worth mentioning that binding forces responsible for attracting the cellulosic chains within/between fibers (intra and inter, respectively) were strongly believed to be hydrogen bonding and Van der Waals forces (Figure 5). These forces have arisen between the negative hydroxyl groups (OH^−^) and H^+^ belonging to the cellulose macromolecule.

### 2.6. Characterization’ s Procedures of the C. Rayon Fibers

Fibrous, mechanical, physical, and chemical properties of the rayon fibers were investigated [52,53,54,55,56,57,58,59,60,61,62,63,64,65,66,67,68,69,70,71,72,73,74,75,76,77,78,79,80,81,82,83,84,85,86]. These studies were conducted based on certain research and/or standard test methods such as ASTM D3379–75 [52] for tensile strength and Young’s modulus for high modulus single filament fibers, BS ISO 11566 [53] for carbon fiber determination of the tensile properties of single filament specimens, ASTM C1557–14 [54] for tensile strength and Young’s modulus of fibers, ASTM D1294 [55] for tensile strength and breaking tenacity of wool fiber bundles 1 in. (25.4 mm) gage length, ASTM C830–00 [56] for apparent porosity, liquid absorption, apparent specific gravity, and bulk density of refractory shapes by vacuum pressure, and ASTM D861–07 [57] for use of the Tex system to designate linear density of fibers, yarn intermediates, and yarns, PA, USA. The calculations of these properties are presented in Table 2A–C.

SEM research was used to examine the anatomical structures of the four selected species [33]. A thin wood chip, 0.5 × 0.5 × 0.1 cm, was sputtered in a vacuum chamber with a 15 nm thick gold coating (JEOL JFC-1600 Auto Fine Coater, JEOL, Tokyo, Japan) after being placed on a carbon tape on an Al-stub and left to air dry. An FEI (Tokyo, Japan) Quanta FEG 450 SEM was used to evaluate the materials. In addition, 5 to 20 kV of accelerating voltage was used to operate the microscope.

Furthermore, the crystallinity index of the c. rayon was determined by performing the X-ray diffraction analysis.

Moreover, the thermal properties, namely mass loss via thermogravimetric analysis, heat change via differential thermal analysis, and scanning differential calorimetry (glass transition temperature).

#### 2.6.1. Fibrous Properties

Three of the essential fibrous properties, namely staple length, linear density, and fiber diameter, were investigated, and their evaluation was conducted as illustrated in Table 2A.

The staple length of the c. rayon is the average length of a group of fibers based on their origin. In this invention, it was measured by an ordinary ruler for straight fibers that were arranged and fixed on a dark background [44].

The linear density of the c. rayon is a measure of the mass per unit length of a fiber that controls the fiber’s fineness. The Tex system was used in this study to estimate the linear density of the rayon, according to [57]. In the Tex system, the weight of fibers was recorded in grams for a 1000 m length based on the equation explained in Table 2A [58].

In addition, SEM study was used to study morphology and measure the fiber diameter of the c. rayon using the Field Emission SEM device (JEOL, JSM-7600F). The accelerating voltage was set to 5 kV. Loose fibers were glued onto aluminum stubs using carbon tabs. Prior to examinations, all specimens were sputtered with a 15 nm thick gold layer (JEOL JFC-1600 Auto Fine Coater) in a vacuum chamber to increase the electric conductivity of the rayon fibers in order to enhance the image resolution [2].

#### 2.6.2. Mechanical Properties

Measuring the modulus and strength by conducting tensile tests on a strand of fibers is very popular in the fiber manufacturing industry. Different tensile properties of the rayon fibers, namely, tensile strength (TS), modulus of elasticity (MOE), elongation at break (EB), and breaking tenacity (BT) were investigated based on standard test methods and related research guidelines [52,53,54,59,60,61,62,63,64,65,66,67]. The tensile properties were measured using Instron model 1193 Instron Co., Ltd., Canton, OH, USA. with a 200 N load cell (Appendix A). The machine was calibrated, and the system compliance (Cs) was determined before testing the specimens, as indicated by Ilankeeran et al. [60]. The calculation of the mechanical properties of the c. rayon fibers is illustrated in Table 2B.

##### Tensile Strength

Tensile strength is one of the important properties for describing the mechanical performance of rayon fibers. A universal testing machine was used to determine the tensile strength of the test specimen [68].

The maximum tensile stress (σ_t_) that fiber can endure this load [69] was determined by dividing the peak force by the cross-sectional area of a plane that is perpendicular to the fiber axis, either at the fracture spot or in its immediate vicinity [53,54,60]. The tensile test was performed on single filaments of the rayon fiber [52,53]. The Instron machine is used with a 200 N load cell. The cross-sectional area of a single fiber (Appendix A) is calculated using the mean diameter determined from the SEM images using its scale bar (Figure 2).

Fiber that was randomly extracted from a bundle or spool was inserted into the testing equipment and subjected to a steady rate of displacement until it failed under stress. A valid test result is one where fiber failure does not occur within the gripping region. The specimen’s overall length should be adequately extended (at least 1.5 times the length of the gage) to facilitate easy handling and gripping [53,54].

The individual fibers were affixed to a paper frame measuring 3 cm × 2 cm using different lengths of gauge. The gauge lengths were adjusted to 15, 25, and 30 mm. An appropriate adhesive was utilized to secure the fibers in place on a paper frame. The individual fiber test specimens were thereafter inserted into the testing frame, which was equipped and loaded with a 200 N load cell at a speed of 1 mm/minute until it reached the point of failure. Approximately five specimens were made and tested for each indicated property. The Young’s modulus (MOE) of the fibers was adjusted by subtracting the compliance (C) of the machine from the total stiffness obtained from the load vs. displacement data [53]. The tensile strength was calculated using the formula presented in Table 2B [54].

##### Modulus of Elasticity (MOE)

The MOE, known as Young’s modulus or elastic modulus, is determined from the linear region of the stress–strain curve.

The modulus of elasticity (MOE) of the fibers was adjusted by subtracting the compliance (C) of the machine from the total stiffness obtained from the load vs. displacement graph, as described by Anonymous [53] using the equation provided in Table 2B. Each gauge length was examined using five specimens. The provided results represent the mean values, while the errors indicate the standard deviation from the mean [70].

##### Elongation at Break (EB)

The EB is an important fibrous property that directly affects yarn quality, especially elongation, toughness, or work-to-break values [71,72,73,74,75,76,77]. It is the increase in length beginning from the beginning of the stress–strain curve to the point corresponding with the breaking force (Table 2B).

Measurement of the apparent elongation was measured and recorded for the acceptable specimens at the breaking force. The average of the breaking force observed for all succeeded specimens, where the ultimate force exerted on the specimen was read directly from the testing machine, was recorded. The apparent elongation as the percentage increases in length is based on the gage length (initial nominal testing length of the specimen (Table 2B).

##### Breaking Tenacity (BT)

The BT is the breaking strength of fibers (BT) or the maximum load that can be supported by the fiber. The determination was based on the standard test method for tensile strength and breaking tenacity of wool fiber bundles’ one-inch gage length: ASTM, D1294 [55]. Breaking tenacity is measured in gram force per Tex. It was determined based on the calculations shown in Table 2B.

#### 2.6.3. The Physical Properties [78,79,80,81,82,83,84,85,86,87,88,89,90,91,92,93,94,95,96,97,98,99,100,101,102,103,104,105,106,107,108,109,110,111,112,113]

##### The Cellulose Yield (α–CY)

For the α–CY of the wastepaper, the productivity of the α–cellulose due to its isolation from the wastepaper via the hydromechanical system can be estimated using what is presented in Table 2C [16].

##### The Rayon Yield (RY)

Regarding the RY, it is worth mentioning that after curing the rayon fibers within the hardening bath containing citric acid (10%, wt/wt), the fibers were transferred to wash, air-dried, and oven-dried at 70 °C for 2 h. It was calculated based on the formula shown in Table 2C [16].

##### The Apparent Density (AD) of the C. Rayon

The AD of the c. rayon was obtained by following the principle of Archimedes (the buoyancy method) according to the ASTM C830-00 [56]. Initially, each rayon fiber specimen was dried in a vacuum oven for 2 h, then it was immersed in water in a vacuum descant until complete saturation was reached and no air bubbles were released. The fibers were suspended in water and weighed (S). The AD was calculated by using the equation presented in Table 2C [79].

##### The Moisture Content (MC) of the C. Rayon

The MC of the c. rayon is defined as the weight of water in the rayon fibers expressed as a percentage of the total weight, as indicated in Table 2C [16,79].

##### The Moisture Regain (MR) of the C. Rayon

The MR of the c. rayon is the moisture absorbed by a particular weight of the rayon fibers in the standard atmosphere after drying at 105 ± 3 °C for one hour until obtaining constant weight. The MR was calculated by immersing a known weight of oven-dried rayon fibers (W_2_) in 100 mL of deionized water at 25 °C for a certain time until no excess water was absorbed by the fibers and weighed (W1). The MR was determined according to Kandemir et al. [79] and Zaman and Begum [16] as clear from Table 2C.

##### The Volumetric Shrinkage (VS) of the C. Rayon

The VS of the c. rayon was calculated by the formula shown in Table 2C [59].

#### 2.6.4. The Spectroscopic Analysis of the C. Rayon

##### The FTIR

The chemical structure (functional groups) of the c. rayon material was investigated using a Bruker Tensor 37 FTIR spectrophotometer, Toronto, ON, Canada. The samples were mixed with KBr at a 1:200 (wt/wt) ratio and compressed under vacuum to form pellets after being oven-dried for about 4–5 h at 100 °C. The transmittance mode was used to record the samples’ FTIR spectra [56,57], which covered a range of 4000–500 cm^−1^.

##### The X-ray Powder Diffraction (XRD)

The XRD spectra of the c. rayon were used to examine the crystallinity of the fibers using an XRD 7000 Shimadzu diffractometer (Kyoto, Japan). The device consists of a rotating anode generator with a copper target and a wide-angle powder goniometer. The generator was operated at 30 KV and 30 mA. For three milliseconds, the samples were subjected to CuKa radiation at a wavelength of 0.15418 nm. All experiments were conducted in reflection mode, with scan speeds of 4°/min and increments of 0.05°. Each sample was scanned between 4° and 30° in the range of 2θ = 26°. Initially, the individual crystalline peaks for the crystallinity index were extracted from the diffraction intensity profiles using a curve-fitting approach [47,111,112]. 

The CI was calculated by dividing the area of the crystalline cellulose diffractogram by the entire area of the diffractogram. Hindi indicates that the area under the curve was determined by adding surrounding trapezoids in Excel (Microsoft, USA).

The intensity in a.u. in relation to the two-theta, given the crystallographic planes of 110-, 110, and 200, was determined. Crystallinity Index (CI) was calculated from the XRD diagrams [47,111,112].

#### 2.6.5. Thermal Analysis of the C. Rayon

Thermogravimetric analysis (TGA) is a method of thermal analysis in which the mass of a sample is measured over time as the temperature changes. This measurement provides information about physical phenomena, such as phase transitions, absorption, and desorption, as well as chemical phenomena, including thermal decomposition and solid-gas reactions (e.g., oxidation or reduction).

The term phase transition (or phase change) is most used to describe transitions between the solid, liquid, and gaseous states. In addition, a decomposition reaction is a thermally induced reaction of a chemical compound forming solid and/or gaseous products. Furthermore, solid-gas reactions are a type of heterogeneous solid-state reaction occurring when a reactive solid is exposed to a stream of reactive gas. Typical examples of solid-gas reactions are sorption and corrosion of metals. Moreover, oxidation can describe different processes in the context of thermal analysis.

An analysis of the thermal properties of the c. rayon, besides its cellulosic precursors (crude and purified wastepaper), was performed to monitor its processing temperature range and its endurance. Two thermal analyses were investigated, namely thermogravimetric analyses (TGA) and differential thermal analysis (DTA) using a Seiko & Star 6300 analyzer, EXSTAR, TG/DTA 6300, Seiko, Japan, Central Laboratory, Faculty of Science, Alexandria University, Egypt. Heating was scanned from 30 °C up to 500 °C under a heating rate of 20 °C/min in a flowing nitrogen gas [81,82,83,84,85,86,87]. The mass loss of the samples was estimated from the TGA curve () using the following equation: Mass loss = [(W_2_ − W_1_)/W_1_] × 100, where W_1_ = Initial sample weight for such a temperature region. W_2_ = Final sample weight for the same temperature region.

Heat change (μVs/mg) of the c. rayon was estimated upon thermal exposure from 30 °C up to 500 °C under a stream of an inert atmosphere of N_2_ via the differential thermal analysis (Seiko & Star 6300 thermo-analyzer). The estimation process was performed using the DTA thermograms using the STARe excellence thermal analysis software.

#### 2.6.6. Chemical Properties of the Rayon Fibers

The viscosities of the cuoxam solution containing the cellulose extracted from the wastepaper precursor were measured using an Ostwald viscometer. Accordingly, the molecular weight (MW) was calculated using the formula presented in Table 2C [59,87]. The degree of polymerization (DP) of the c. rayon is the number of the 4-1-β-D-glucopyranose monomers constituting the rayon. According to Javed et al. [88], it was determined as indicated in Table 2E.

### 2.7. Statistical Design and Analysis

Various properties of the c. rayon were assessed using a randomized complete block design [89], and the statistical analysis of the obtained data was carried out using the analysis of variance approach (ANOVA) and the least significant difference test (LSD) at 0.05.

## 3. Results and Discussion

The output data obtained from the statistical analyses are presented in Table 3 and Figure 5, Figure 6, Figure 7, Figure 8, Figure 9, Figure 10 and Figure 11. All the findings were extensively illustrated and discussed below:

### 3.1. The Scientific Illustration of the Rayon Formation

Regarding its basic structure (Figure 5), cellulose is a linear syndiotactic homopolymer composed of D–anhydro–glucopyranose units (AGU) that are connected by β-(1-4)-glycosidic bonds [1,8,9,90,91]. These chains are aggregated to form microfibrils with cross dimensions ranging from 2 to 20 nm [3]. The size of the cellulose molecule can be defined by its average degree of polymerization (DP). The average molecular weight is estimated from the product of the DP and the molecular mass of a single AGU. Each AGU bears three hydroxyl groups (one primary and two secondary moieties that represent more than 30% by weight), except for the terminal ones. These structural features make cellulose surface chemistry quite intriguing and open a broad spectrum of potential reactions, which typically occur in the primary and secondary hydroxyl groups [6].

The presence of hydrogen bonds with an energy of up to 25.0 kJ/mol in cellulose strongly reduces the range of solution treatments available for it. Accordingly, searching for a new solvent for cellulose must be restricted in at least two directions: (a) The 1st solution must be capable of forming hydrogen bonds with an energy above 25 kJ/mol. In nonaqueous mixed solution systems, the solubility of cellulose is controlled by the number of reactive cations formed and by the polymer preactivation conditions. Another option is to decrease the energy of hydrogen bonds in cellulose by physicochemical or chemical modification [5].

Deeply speculating Figure 5d revealed that some cellulose vapors were emitted from the hot surface of the sample under vacuum. These ideal conditions for softening of the c. rayon were dominated in the samples’ chamber of the SEM device. Accordingly, these scarce images were obtained coincidently upon higher focusing imaging of the fibrous c. rayon structure since increasing focusing requires rising the dominating voltages that are responsible for elevating temperature around the sample upon imaging.

Suggestions for the mechanism of cellulose dissolution are reviewed below:

The cuoxam solution, known as cuoxam reagent, is a metal ammine complex having the formula [Cu(NH_3_)_4_(H_2_O)_2_](OH)_2_] as shown in Figure 5. It is featured by its deep blue aspect (turquoise color) and is useful for dissolving cellulose. It was reported by Burchard et al. [85] that Schweizer, in 1857, succeeded in effectively dissolving cotton using copper salts and concentrated ammonia.

Figure 5 clearly demonstrates that cuoxam forms coordination bonds with the deprotonated hydroxyl groups located at the C_2_ and C_3_ locations of the anhydro–glucopyranose units (AGU). It is evident that the hydrogen binding between the primary OH group in the C6 position and the ring oxygen of the following AGU is disrupted while the other hydrogen bond is strengthened [91]. The presence of these bonds on both sides of the chain results in increased chain rigidity [92,93]. This proposition was recently substantiated through systematic investigations of crystal structures involving oligosaccharides in conjunction with computer simulations and calculations of complex stability constants [16,93], as well as light scattering measurements [91,92].

The regenerated brittle cellulose was produced through the reaction of a copper–ammonia complex with citric acid (10% wt/wt), as described by Zaman and Begum [16]. The existence of one primary (C_6_) and two secondary (C_2_, C_3_) hydroxyl groups in an anhydro-glucose unit of cellulose dictates the occurrence of a system of inter- and intrachain hydrogen bonds in the native polymer, as shown in Figure 5. This significantly restricts the variety of single-component solutions that are acceptable for practical applications. Analyzing the IR absorption spectra of cellulose in the OH stretching region (3000–3700 cm^–1^) allowed for the clarification of the organization of hydrogen bonds within and between chains.

The theoretical estimates confirmed the occurrence of two intra- and one interchain hydrogen bonds in the elementary unit of native cellulose. The hydrogen bonding energy in cellulose ranges from 8.5 to 15.0 kJ/mol. Dissolution of cellulose requires the breaking of at least all the interchain hydrogen bonds. There are several solutions capable of forming hydrogen bonds with the energies Eh above 13.0 kJ mol. It was estimated that the hydrogen bonding energy to be Eh = 25.0 kJ/mol [87,88], which is about twice that published in the literature for interchain hydrogen bonds in cellulose [5].

### 3.2. Characterization of the C. Rayon Fibers

#### 3.2.1. Fibrous Properties of the C. Rayon

Three fibrous properties, namely staple length, linear density, and fiber diameter of rayon were investigated in this study, and their mean values are presented in Table 3.

##### The Staple Length (SL)

The SL property was measured, and the data were presented in Table 3. The mean SL value was found to be about 44 mm, which is longer than that produced by Abdul Basit et al. [66] and lies within the ordinary limit. The average SL of a group of fibers was reported to be dependent on the origin of the fibers. Increasing the average staple length of the fibers can ultimately make the yarn softer [44]. Accordingly, the invented rayon fibers were better than those produced by Abdul Basit et al. [66].

The rayon fibers can be produced in extremely fine deniers to obtain softness and handle characteristics similar to silk. The burning characteristic of this material was reported to be similar to viscose rayon, whereby it burns rapidly and chars at 181 °C. However, this temperature was higher than that referred to by Kotek [33].

##### The Linear Density (LD)

The LD property is the mass per unit length and was measured in the Tex unit (weight in grams per kilometer). It was used to characterize the LD of the c. rayon fibers [57]. The average of the SL was about 235 Tex, which is higher than the range determined by Aytac et al. [23] and lower than that found by Shamsuddin et al. [70], as indicated in Table 3. Since the LD is used as a measure of fineness, it can be said that the lower the LD value, the more fineness for the rayon fibers. Accordingly, the invented rayon fibers had lower fineness than those synthesized by Shamsuddin et al. [70]. It is worth mentioning that the LD can be adjusted easily by controlling several parameters, especially the diameter of the syringe’s nozzles, the viscosity of the cuoxam/cellulose solution, and the pressure of the extruding force. For more illustration, reducing the diameter of the syringe’s nozzles, reducing the viscosity of the cuoxam/cellulose solution, and increasing the extruding force can help to idealize the rayon fineness.

##### The Fiber Diameter (FD)

The FD character is an additional indicator of rayon fineness besides the LD. The FD value was estimated from the SEM image using its scale bar (Figure 3) and was found to be about 19.4 µm. The obtained average is lower than that found in another study (54 µm) conducted by Aytac et al. [23], as is clearly seen in Table 3.

It is worth for mention that the original rayon fibers presently being produced have circular shape of fibers, but the wrinkles shown in Figure 3 can be attributed to the volumetric shrinkage of the fibers upon excessive drying occurred due to the fiber exposure to the high voltage in the specimen chamber of the SEM device. Smaller FD values allow yarns to be more even, lustrous, and stronger [16].

#### 3.2.2. Mechanical Properties of the Cuprammonium Rayon

The results were analyzed and presented in Table 3 and Figure 6, Figure 7, Figure 8, Figure 9, Figure 10, Figure 11 and Figure 12.

##### The Tensile Strength (TS)

The TS average of the rayon is 218.3 MPa, as indicated in Table 3. The lower strength of the rayon fibers obtained in the present study compared to that for regenerated cellulose (360 MPa) fabricated by Dirgar [99] can be attributed to the lower crystallinity and higher amorphous regions in the rayon structure and vice versa for Tencel material [103,104,105]. Changes in tensile strength and elongation with wetting may depend mainly on the number of molecular chain ends in the amorphous region [59].

The modulus of elasticity is a very important characteristic in handling rayon yarns during such weaving and stentering when sudden tensions are applied [59]. The MOE of the rayon fibers was determined to be 14.3 GPa (Table 3). This average is close to the estimated range of rayon material (15–15.6 GPa) found by Seavey and Glasser [105] and Shamsuddin et al. [70].

The mean average of elongation at break estimated for the rayon fibers invented is about 16.1% (Table 3), which is a median of the global range. Ordinary viscose rayon has 10.6–36.8% elongation at break at break as determined previously by Miyake et al. [59], Aytac et al. [23], Iqbal and Ahmad [98], and Abdul Basit et al. [66].

The breaking tenacity of the resulting rayon fibers had a high mean value (21.37%) as compared to that referred by Miyake et al. [59], whereby high tenacity rayon fiber has about 9–17% value. Furthermore, our BT average lies within the determined range (4.42–58 cN/Tex) found in the literature [66,98,107], as presented in Table 3.

### 3.3. Mechanical Properties of C. Rayon as Affected by Some Processing Parameters

Presenting the influence of the mechanical properties of the c. rayon by the most important processing parameters is clear in Figure 7, Figure 8, Figure 9, Figure 10, Figure 11 and Figure 12.

Regarding the ammonia’s injection rate upon synthesis of the cuoxam reagent, it is obvious from Figure 7 that the TS increased significantly from 179.2 MPa to 218.3 MPa when the injection rate was raised from 60 to 120 mL/minute. Furthermore, there was no significant change in the tensile strength trend above the injection rate of 120 mL/min.

#### 3.3.1. Ammonia’s Injection Rate

It is worth mentioning that the same trend recorded for the TS was repeated for the other tensile properties studied (MOE, EB, BT). As a further illustration, we note that the injection rate of 120 mL/min. was revealed to be the highest mean value of the MOE, EB, and BT (14.3 GPa, 16.1%, and 27.35%, respectively), as indicated in Figure 8.

Accordingly, the best injection rate of the ammonia gas for producing the cuoxam reagent in the present invention was found to be 120 mL/minute because of the enhancement that occurred for all the mechanical properties examined (Figure 7 and Figure 8).

#### 3.3.2. Ammonia’s Injection Duration

Extending to studying the influence of the mechanical properties of the c. rayon by the processing parameters, it can be seen from Figure 9 that injecting NH_3_ gas for 6 min gave the highest tensile strength’ mean value (215.93 MPa). In addition, there were no statistical differences between the following durations (9 and 12 min). On the other hand, injecting the ammonia gas within the rayon’s synthesis vessel for six minutes did not achieve the target tensile strength quality, given the lowest tensile strength value.

Regarding the remaining mechanical properties (MOE, EB, and BT) investigated for the c. rayon fibers, it was observed that they reached their maximum values at the injection duration of six minutes (Figure 10).

These findings for all the four mechanical properties determined (TS, MOE, EB, and BT) can be explained by the inadequate amount of the NH_3_ allowed to saturate the permeable cellulose within this short duration by higher quantities of cuoxam solvent enhancing the rayon production (Figure 9 and Figure 10).

#### 3.3.3. Hardening Period of the C. Rayon

Studying the four mechanical properties (TS, MOE, EB, and BT) of the c. rayon fibers as affected by the hardening period (Figure 11 and Figure 12), it can be seen from Figure 11 that the hardening period of 20 min revealed the highest TS’ mean value (228.09 MPa). In addition, there were no statistical differences between the following durations of 15, 20, and 25 min (228.09 MPa, 228.09 MPa, and 215.5 MPa, respectively). On the other hand, allowing the recent c. rayon’s fibers to be hardened for 10 min gave the lowest TS value (184 MPa).

An examination of Figure 12 reveals that there are similarities between the resulting trends of the remaining mechanical characteristics (MOE, EB, and BT) and those obtained for the tensile strength.

First, for the MOE, the period allowed for the c. rayon to be hardened was recorded to be 20 min (15.84 GPa), which is significantly similar to those mean values obtained for the other hardening periods of 15 min and 25 min (14.3 GPa, and 13.8 GPa, respectively). On the other hand, when the hardening period was performed during the 10 min, the MOE was the lowest one (11.2 GPa).

Secondly, concerning the EB, the same trend observed for the MOE was recorded for the EB property in which the hardening period of 20 min gave the highest EB value (17.67%). However, the latest value was found to be similar to those resulting in both hardening periods of 15 min. and 25 min. (16.1% and 15.9%, respectively).

Finally, regarding the BT, the hardening period of 20 min achieved the best quality (29.03%) compared to that produced using the short hardening period of 10 minutes (19.25%).

Based on this finding, and upon injecting the ammonia gas through the α–cellulose saturated and immersed in the Cu(OH)_2_ to complete producing the cuoxam solvent, we find that the injection rate of 120 mL/min for obtaining the highest TS for the final product of the c. rayon is preferable. Utilization of higher rates will consume more amounts of the ammonia gas without gaining noticeable enhancement in the c. rayon’s mechanical quality.

#### 3.3.4. Physical Properties

The traits studied were the yields of α–cellulose isolated from the wastepaper as well as the physical properties of the rayon synthesized from the α–cellulose, namely, α–cellulose’s yield (αCY), rayon yield (RY), apparent density (AD), moisture content (MC), moisture regain (MR), volumetric shrinkage (VS), crystallinity index estimated from the X-ray diffractograms (XRD), mass loss determined from the thermogravimetric curves, heat change calculated from the differential thermograms, and the glass transition temperature estimated from the differential scanning calorimetry output as shown in Table 3.

The average of the α–cellulose yield obtained from the wastepaper after the elimination of impurities such as calcium carbonate, gums, and inks was found to be about 90.3%. This productivity was higher than that found previously by Hindi and Abohassan [2], as clearly seen in Table 3. This result reflects the efficiency of the hydromechanical method along with the CaCO_3_–elimination reagent in isolating the cellulose from the wastepaper.

The mean value of RY produced from the isolated cellulose was about 92.25%. This output was found to be higher than the RY range (72.47–88.27%) obtained by other researchers, such as Zaman and Begum [16], as clearly seen in Table 3. Accordingly, it can be said that the higher the RY, the more profit can be gained.

The AD of the rayon fibers was found to be about 1.54 g/cm^3^. Comparing it with that reported by Shishir [97], we see that it lies within the standard limits, as is obvious from Table 3.

As indicated in Table 3, the average MC of the rayon fibers was found to be 8.6%. This was lower than the MC range (10–12.75%) determined by other researchers [16,66].

The MR values of cotton and regenerated cellulose are about 7.8% (Table 3). Due to its cellulosic nature, viscose has more amorphous regions, which give more voids in its structure [109]. This nanostructure of the rayon fibers has some moisture management properties among all viscose blends. It was found to have moisture absorption ability, but it wicks less than natural fibers such as cotton fibers [66,107].

The mean value of the rayon fibers was determined to be about 1.8%, which lies within the accepted region. This value is higher to some extent than that found by Nawaz et al. [90], as presented in Table 3. Since volumetric shrinkage is the dimensional change resulting in a decrease in the length or width of a rayon fiber specimen due to losing some of its moisture content, their fabrics’ dimensions are expected to decrease slightly when exposed to heat or evaporation conditions. However, this VS was small enough to be undistorted for the final product fabrics of the rayon fibers. Since rayon fibers are composed mainly of cellulose, which is well known to be hydrophilic in nature, they can allow the water to soak into the fiber and swell as well as lose moisture and shrink. On the other hand, hydrophobic fibers such as cellulose triacetate will exhibit very little shrinkage [108].

#### 3.3.5. The Spectroscopic Analysis of the C. Rayon

##### FTIR

Concerning the essential functional group and the famous atomic linkages featuring the c. rayon as well as pure cellulose, six absorption bands were detected from the FTIR analysis at 1050, 1285, 1587, 1656, 2852, and 3366 cm^−1^.

The scientific interpretation of the arisen bands can be summarized as follows: (1) At 1052 cm^−1^, the reason for band appearance was due to the stretching band of the C–C ring and the C–O–C glycosidic ether linkage [49,50,51,52]. Furthermore, at 1285 cm^−1^, the scissoring motion of the CH_2_ group was responsible for the appearance of the clear band [49,50,51,52,53]. In addition, the O–H bending of the absorbed water is the reason for the band noticed at 1587 cm^−1^ [49,50,51,52,53]. Moreover, the C–O stretching vibration for the acetyl and ester linkages was believed to be the cause of the band detected at 1656 cm^−1^ [49]. The C–H stretching effect was referred to its relation to detecting the FTIR’s band at 2852 cm^−1^ [49,54,55]. Finally, the band observed at 3366 cm^−1^ was correlated to the effect of O–H stretching (axial vibration) intramolecular hydrogen bonds [56,57].

Based on the spectral FTIR data, and since the detected bands are adopted to those found by several researchers [114,115], it is proven that cellulose makes up the whole c. rayon fibers’ backbone. Accordingly, the novel regenerating process of the alpha cellulose isolated from wastepaper attained the virgin chemical constitution of the c. rayon produced.

##### XRD

The CI value obtained approaches those found for rayon (71.62%) by Smole et al. [110], cellulose (76.01%) by Wulandari et al. [111], wood pine (70%) by Borysiak and Doczekalska [60] fall between the ranges of 56% to 78% determined by Terinte et al. [111] and 41.5% to 95.5% shown by Park et al. [112]. These results were obtained using various cellulosic precursors and different measuring techniques. Furthermore, the present CI results are consistent with those reported by Kumar et al. [113], who found that the removal of lignin and hemicelluloses as amorphous parts during acid hydrolysis resulted in a bagasse CI range of 35.6% to 63.5%.

Investigating effect of the crystallite size of the c. rayon on its performance, it is commonly recognized that the optical, chemical, and physical characteristics of cellulosic fibers may be strongly associated with their crystallite orientation [116]. Therefore, the rules controlling the crystallite orientation in synthetic and semi-synthetic fibers are both theoretically and practically significant. For instance, when the orientation of rayon fibers is changed from a random to a parallel orientation, the tensile strength and elongation may vary by several hundred percent. Stretching is the typical technique used to increase orientation in rayons. As a result, the main axes of cellulose crystallites will be aligned parallel to the stretching’s direction [116].

Furthermore, when swollen cellulose is dehydrated, the minor (secondary) axis “101-plane” of the crystallite is orientated parallel to the shrinkage’s direction. In order to study the factors that affect the production of orientation both individually and collectively, as well as to formulate the general orientation behavior of cellulose into a rule, the current paper aims to expand on previous investigations on the orientation behavior of the native cellulose crystallite (2, 3) to that of the hydrate cellulose crystallite, which exists in fibers produced from coagulated cellulose [116].

The crystallinity index estimated from the XRD diffractograms of the three cellulosic resources is shown in Figure 13a. Furthermore, the crystallinity indices are affected by the most important parameters of the gaseous ammoniation process, namely ammonia’s injection rate (Figure 13b), ammonia’s injection duration (Figure 13c), and hardening duration (Figure 13d).

It can be seen from Figure 13a that the CI value was increased from wastepaper up to cellulose. In between these, the c. rayon material occupied a lower level than the α-cellulose. This finding can be attributed to the higher crystallinity of the α-cellulose, whereas the lowest CI estimated for wastepaper was related to its various natural/synthetic/additives that blended to the virgin cellulose used to fabricate the original writing paper, the α-cellulosic precursor of the c. rayon in the current study [47,90]. Moreover, it was found that the c. rayon material had a CI value lower than that for the α-cellulose itself.

In addition, for the three above-mentioned parameters of the gaseous ammoniation process, it was found that the ammonia’s injection rate (Figure 13b), the injection rate of 120 min produced the c. rayon with the highest CI. On the other hand, the injection rate of 240 min gave the lowest CI value. In between, the injection duration of 60 and 180 min lies in a median situation. This finding can be related to that.

Furthermore, for ammonia’s injection duration (Figure 13c), the highest CI’s mean value was achieved by using the 6 min, while the lowest one was obtained using the 3 min. In between, the injection durations of 9 and 12 min gave median CI features.

Moreover, performing the hardening duration of 20 min gave the highest CI value (Figure 13d). In contrast, the period of 25 min gave the lowest CI value for the c. rayon. Investigating the median hardening graphs (10 and 15 min), we note that they represent average situations between those representing the 20 and 25 min’ graphs.

#### 3.3.6. Thermal Analysis

##### Thermogravimetric Analysis (TGA)

TGA, the dynamic phenomenological method for examining a material’s thermal characteristics, is used to find out how the sample weight reacts to temperature variations. From this technique, the overall mass loss of the c. rayon materials occurred upon heating ranged from 30 °C to 500 °C (the sum reduction in weight via four temperature regimes, namely 30 °C–200 °C, 200 °C–300 °C, 300 °C–400 °C and 400 °C–500 °C). Table 3 revealed that the total mass loss of the c. rayon material equals about 36.27%, which approaches the featuring cellulose and is higher than several synthetic fibers. I addition, this mean value lies within the normal scale of the rayon products found by other researchers.

##### The Differential Thermal Analysis (DTA)

The mean values of the heat change (via the DTA) presented in Table 3 were found to be −884 μVs/mg and +1247 μVs/mg for endo- and exotherm, respectively. However, these data lie within the ordinary output from several research results. This finding indicates that the obtained c. rayon had enough thermal stability to endure the expected trade applications.

##### The Glass Transition Temperature (Tg)

The glass transition temperatures of pure amorphous water and cellulose have been reported in the literature to be about 220 °C [86].

The importance of determining the Tg of the c. rayon and cellulose derivatives is summarized as follows: the temperature at which an amorphous polymer transforms from a hard, glassy form to an elastic, rubber-like state or into a viscous fluid is known as the “glass transition temperature”. The polymer’s mechanical and physical temperature derivatives alter during this secondary transition, known as the glass transition, as indicated by Salmén and Back [86]. The main chain’s degree of movement, such as the rotation of a longer chain segment, is correlated with the glass transition temperature. Numerous hypotheses explain this by relating it to the polymer’s void volume, or the volume that is empty of molecules [86].

It is hypothesized that the void volume is sufficiently high above the Tg in which a large chain motion is conceivable and exists, such as a rotation of segments. This suggests that a polymer, particularly one having side groups, can undergo several secondary transitions, such as when the void volume is adequate to rotate one side group. It is believed that the secondary transitions of cellulose occur at zero degrees for movement in the glucopyranose ring and at 220 °C for the main chain movement, which is the genuine glass transition [80,117,118,119,120,121].

Water has a significant plasticizing effect on cellulosic materials. Water is a natural softening agent for cellulose and is used in nearly all cellulose production, conversion, and finishing processes. This softening is particularly interesting because it allows the stiff fibers in hard fiber building boards to become more flexible during the press-drying process, resulting in a larger bonding area. Another illustration is the corrugating of fluting, in which the roll profile is molded around the paper by the water. However, not much research has been conducted on how water affects cellulose’s glass transition temperature [122].

#### 3.3.7. Chemical Properties

Two chemical properties of the cuoxam solution saturated with cellulose, namely molecular weight and degree of polymerization, were analyzed and are shown in Table 3.

##### The Molecular Weight (MW)

The MW of the cuoxam solution saturated with cellulose is presented in Table 3. The molecular weight of viscose is 90,000 to 110,000. However, the MW can vary widely depending on the degree of polymerization, which refers to the number of repeating glucose units in the cellulose structure.

In general, the molecular weight of cellulose, and thus viscose, can be in the range of 20,000 to 500,000 g/mol or even higher, depending on the source and the processing method used to create the viscose fiber.

##### The Degree of Polymerization (DP)

The average DP of viscose polymer ranges from 300 to 450 Daltons, as indicated in Table 3. Due to the insufficient length of pulp fibers in their original condition, they are dissolved and then reformed into filaments. The rayon fibers produced using the wet spinning method demonstrated that while analyzing various DP values ranging from 290 to 480, DP 430 exhibited the most favorable tensile characteristics [98].

## 4. Conclusions

Rayon fibers were prepared from wastepaper by cuprammonium process due to it being potentially cheap, nonpolluting, easy to handle, and it uses common chemicals. The wastepaper was chemically purified to eliminate calcium carbonate that was added while manufacturing their parent paper to isolate pure cellulose. The cuoxam solution was prepared by using two main precursors, namely copper hydroxide and ammonia gas. The copper hydroxide was produced via reacting copper sulfate (5%, wt/wt) sodium hydroxide (1%, wt/wt) in the ratio of 1:1. The prepared copper hydroxide was exposed to ammonia gas to permit the addition of four ammonia groups on it, producing tetra–ammine copper hydroxide known as cuoxam solution via a novel technique termed as gaseous–ammoniation injection process. The air-dried cellulose was generated by dissolving it in the cuoxam solution. The cellulosic solution was regenerated by extruding it within a hardening bath of citric acid (10%, wt/wt), resulting in rayon fibers having high yield (90.3%) and quality. Physical, chemical, anatomical, and mechanical properties of the resulting cuprammonium rayon were studied and were found to be in the normal global range. The fibrous properties of the rayon fibers, namely staple length, linear density, and fiber diameter, were found to be 44 mm, 235 Tex, and 19.4 µm, respectively. The mechanical properties determined, namely tensile strength, elongation at break, modulus of elasticity, and breaking tenacity, were found to be 218.3 MPa, 14.3 GPa, 16.1%, and 27 cN/Tex, respectively. The fiber characteristics render them favorable for use in making sustainable semisynthetic floss for either insulation purposes or spun threads, woven and nonwoven textile clothing, surface coating, as well as binder for carbon briquettes for fluid purification. Based on this finding, and upon injecting the ammonia gas through the α–cellulose saturated and immersed in the Cu(OH)_2_ to complete producing the cuoxam solvent, we find that using the injection rate of 120 mL/min for obtaining the highest TS for the final product of the c. rayon is preferable. Utilization of a higher rate will consume more amounts of the ammonia gas without gaining noticeable enhancement in the c. rayon’s mechanical quality.

### Future Perspectives

Continuous attempts must be made to address the practical concern of the cuprammonium rayon because of such reasons as toxicity and environmental hazard due to using the ammonia gas, its limited solvency quantitively, solving the problems arising in developing its closed processes. This advice is adapted to those referred by Bochek [5] and Heinze and Koschella [13]. Although the mechanical properties’ values were satisfactory, an enhancement’s target must be considered in the future.

Regarding calcium carbonate and other impurities, such as metals, which may be present in cellulose raw materials, studying how they may affect the stability of the cuprammonium rayon and accurately determining the permissible concentration of each pollutant is very important for future investigations.

In addition, the thermal behavior of rayon is noted, especially the softening trend as well as its being affected by endo- and exothermic, which arises upon differential thermal degradation as a trial to enhance the thermal characteristics of the rayon fibers.

## 5. Patents

A method for isolating alpha cellulose from lignocellulosic materials (US Patent No. 11078624); Method for recovery of cellulosic material from waste lingo–cellulosic material (US Patent No. 11136715); Method for separating lignin from lignocellulosic material (US Patent No. 17407663); Method of obtaining rayon fibers (US Patent No. 11441264).

## Figures and Tables

**Figure 1 polymers-16-02431-f001:**
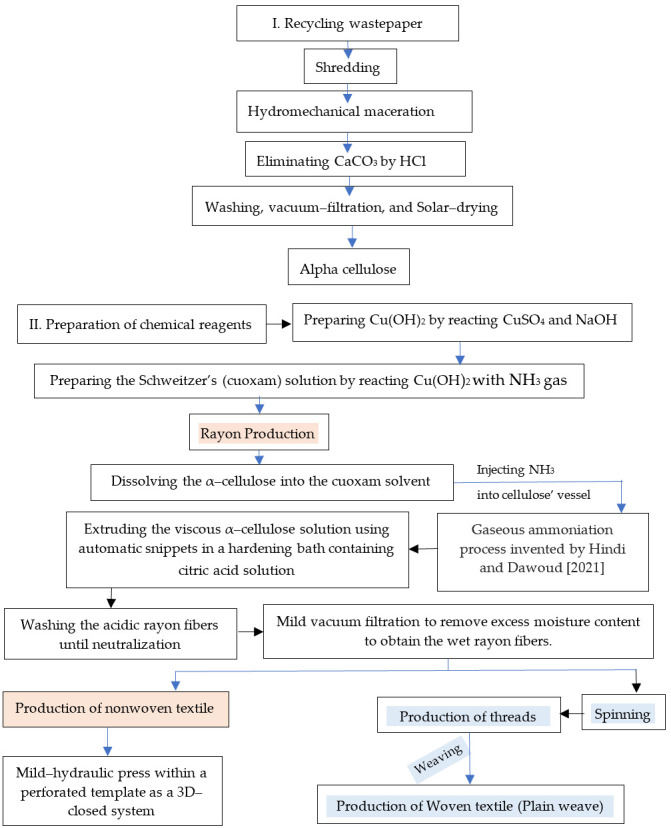
The green practical routes of isolating cellulose from wastepaper and converting it into cuprammonium rayon.

**Figure 2 polymers-16-02431-f002:**
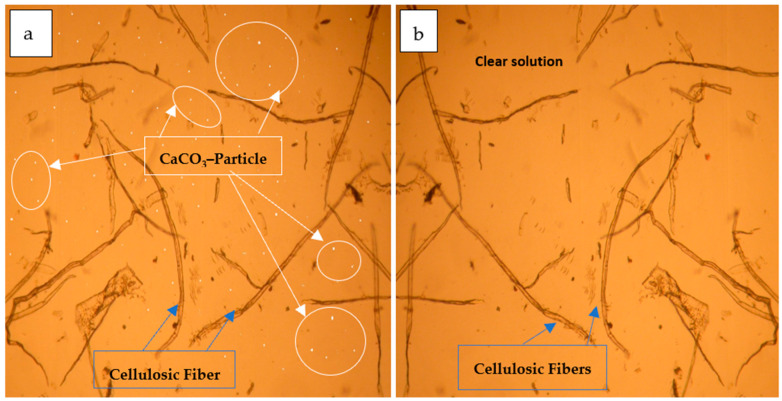
Optical image indicating the dissolution of calcium carbonate particles from the wastepaper using diluted hydrochloric acid (HCl): (**a**) Solution of crude cellulosic fibers; (**b**) Solution of purified fibers after HCl–treatment.

**Figure 3 polymers-16-02431-f003:**
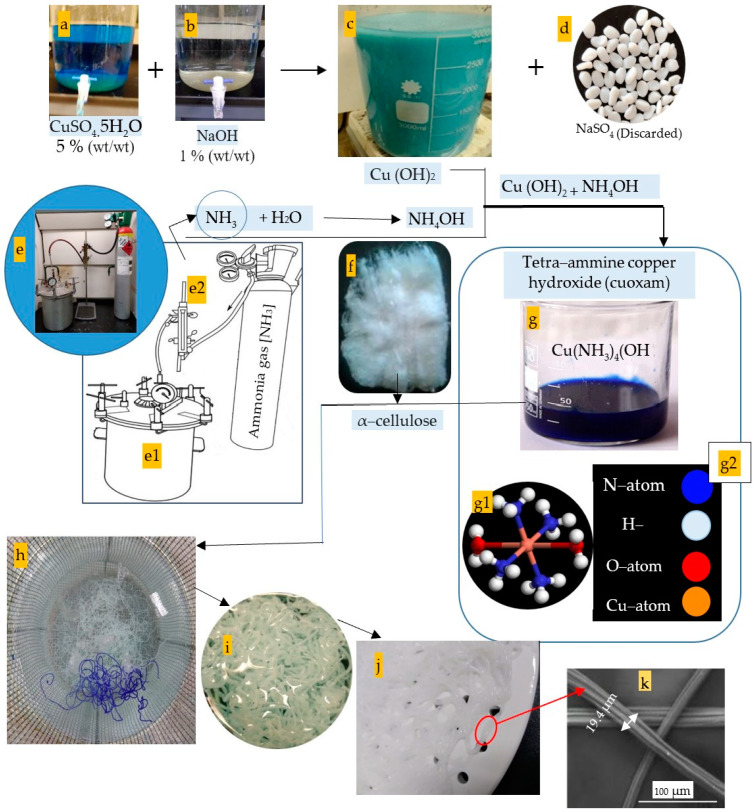
C. rayon fibers’ synthesis: (**a**) CuSO4; (**b**) NaOH; (**c**) Cu (OH)2; (**d**) The discarded NaSO_4_; (**e**) The novel gaseous ammoniation process: (**e1**) The gaseous vessel; (**e2**) Flowmeter; (**f**) α–cellulose precursor; (**g**) The dark blue aspect (turquoise color) of the fresh cuoxam solution (tetra–ammine copper hydroxide): (**g1**) Chemical structure of the cuoxam solution; (**g2**) The atoms’ types constituting the cuoxam solvent; (**h**–**j**) Optical images of the hardened rayon fibers: (**h**) Curing of the rayon fibers within the hardening bath, whereby the blue threads are recently extruded ones, while the white ones are cured that lost their copper ions; (**i**) A close up of cured rayon fibers; (**j**) Air-drying the fabers after neutralization and washing; (**k**) Oven-dried rayon fibers.

**Figure 4 polymers-16-02431-f004:**
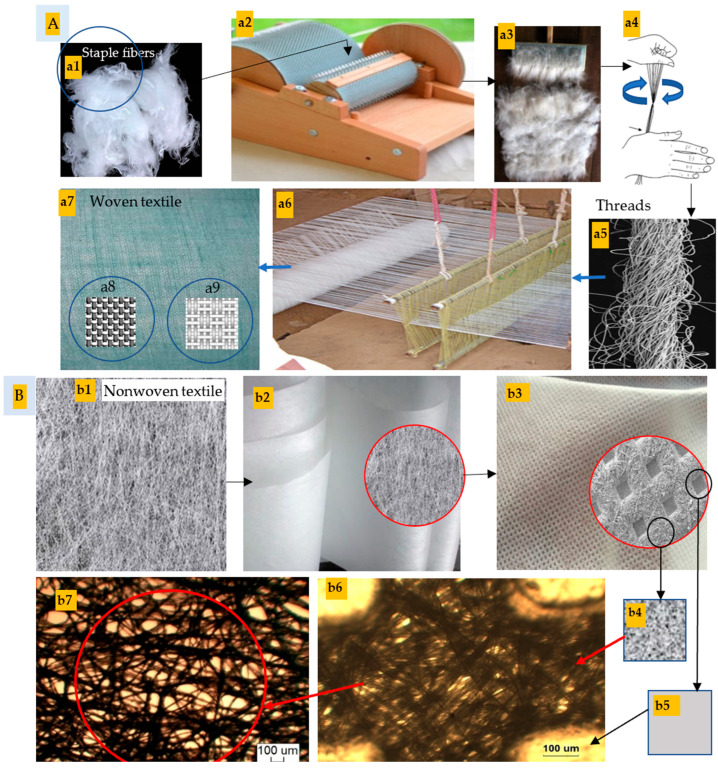
Manufacturing the c. rayon textile from the wastepaper: (**A**) Woven fabric: (**a1**–**a7**) the manual weaving of staple fibers: (**a1**) Staple fibers; (**a2**) Primitive carding machine; (**a3**) Carding brush to obtain ordered and aligned fibers; (**a4**) Spinning the ordered fibers using a hand spindle; (**a5**) Threads; (**a6**) Textile weaving machine; (**a7**) A typical textile; (**a8**) Stretch pattern of plain weave 1/1; (**a9**) Stretch pattern of Panama weave 2/2. (**B**) Nonwoven fabrics: (**b1**) Randomly arranged fibers; (**b2**) Two-dimensional textile sheet; (**b3**) A close-up of a sheet showing a loosened permeable spot (**b4**–**b7**) and a welded spot (**b5**,**b6**) arose upon thermal compression of the randomly arranged fibers.

**Figure 5 polymers-16-02431-f005:**
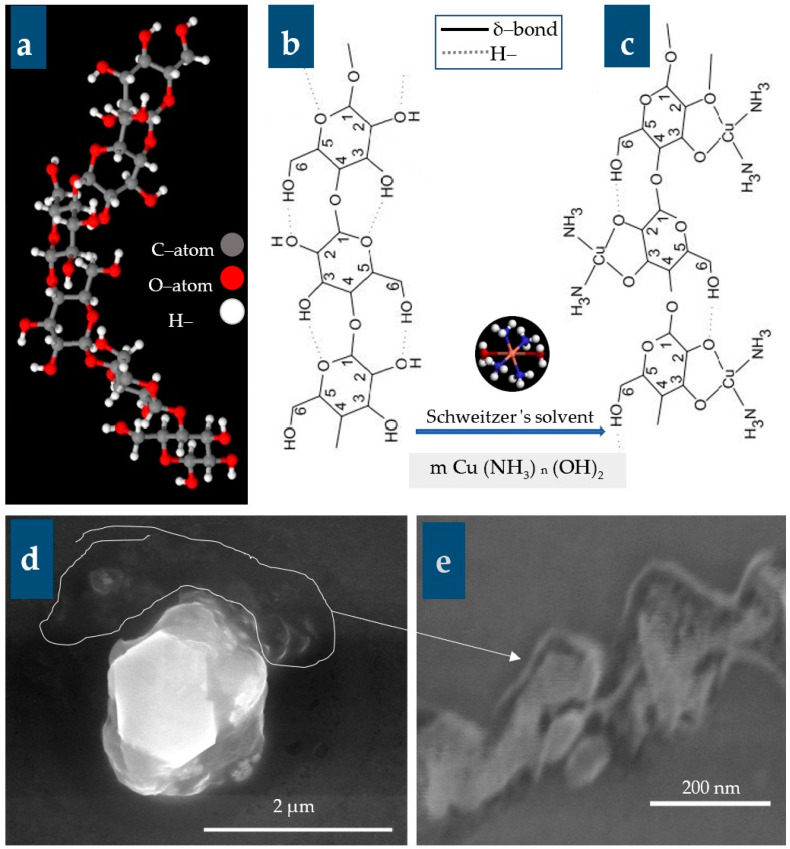
Formation of the cuprammonium rayon polymer: (**a**) Chemical structure of cellulose; (**b**) Part of the hydrogen bonding within cellulose; (**c**) Chemical structure of the regenerated cellulose formed after coordinative binding of copper in cuoxam solution at the deprotonated olate groups in the O_2_^−^ and O_3_^−^ position of the anhydro–glucopyranose unit (AGU); (**d**) SEM images of initially softened–rayon fibers upon exposure to high voltage (20 KV) under vacuum (0.8 mbar) showing the cellulose vapor; (**e**) Close up view of cellulose vapor.

**Figure 6 polymers-16-02431-f006:**
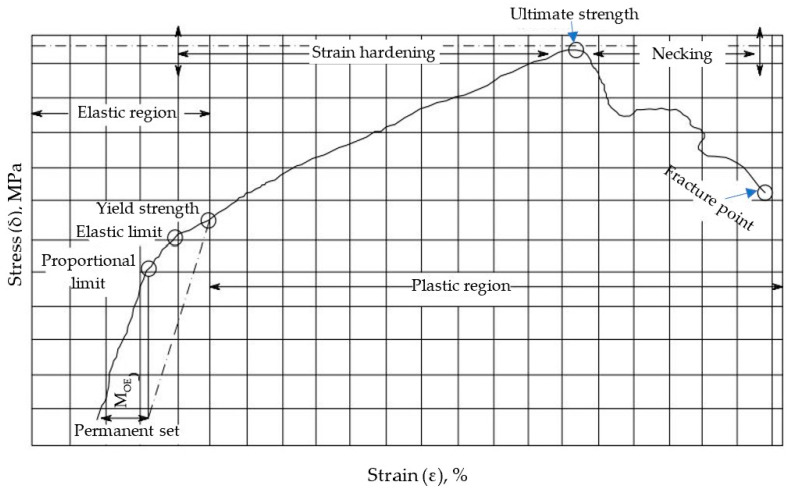
A typical stress (δ)–strain (ε) curve for a single filament of the cuprammonium rayon.

**Figure 7 polymers-16-02431-f007:**
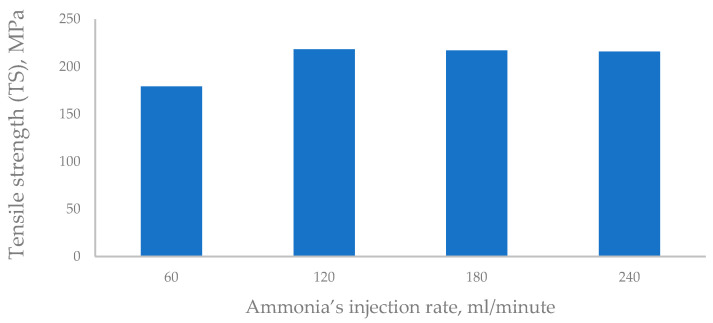
Effect of ammonia’s injection rate on the tensile strength (TS) of the C. rayon produced from the wastepaper.

**Figure 8 polymers-16-02431-f008:**
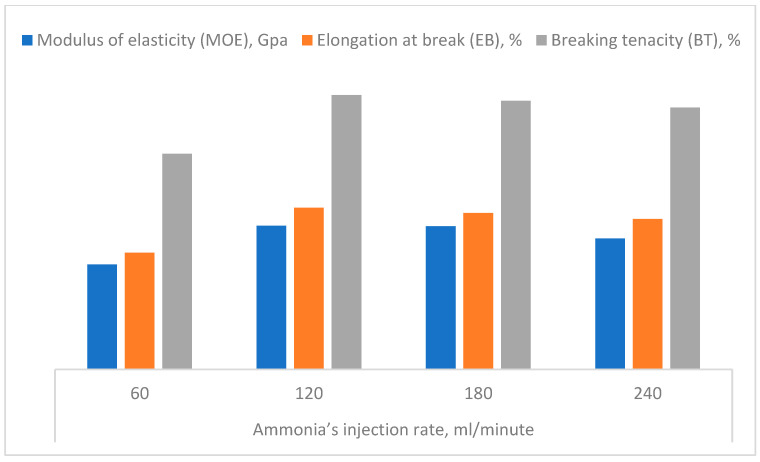
Effect of ammonia’s injection rate on each modulus of elasticity (MOE), elongation at break (EB), and breaking tenacity (BT) of the c. rayon produced from the wastepaper.

**Figure 9 polymers-16-02431-f009:**
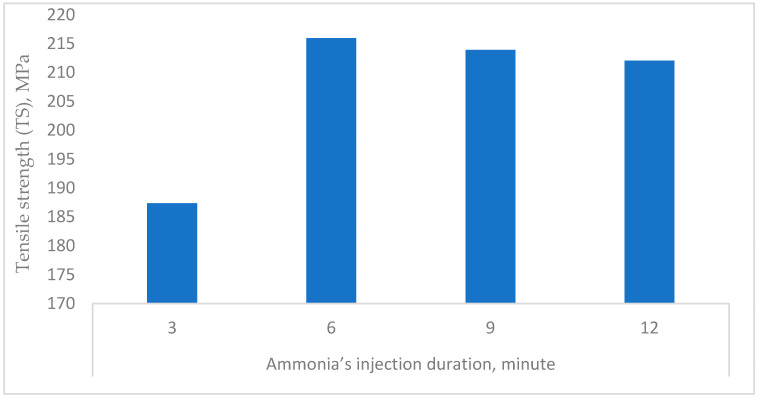
Effect of ammonia’s injection duration on tensile strength (TS) of the c. rayon produced from the wastepaper.

**Figure 10 polymers-16-02431-f010:**
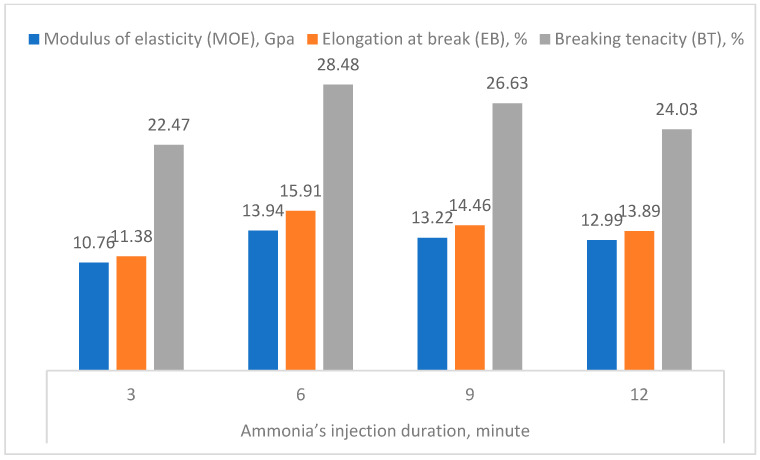
Effect of hardening duration of the c. rayon produced from the wastepaper on its mechanical properties.

**Figure 11 polymers-16-02431-f011:**
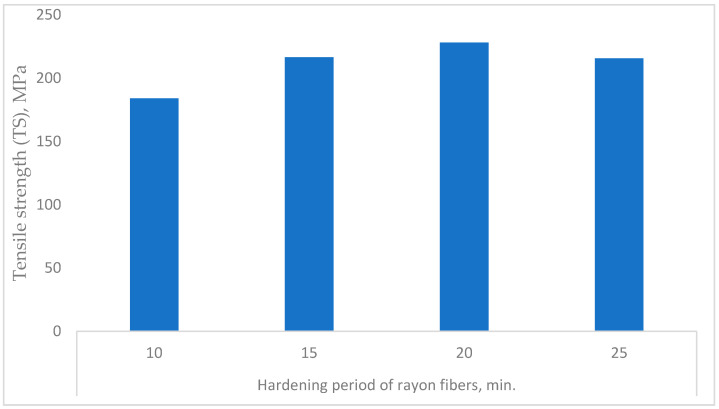
Effect of hardening duration of the c. rayon produced from the wastepaper on its tensile strength (TS).

**Figure 12 polymers-16-02431-f012:**
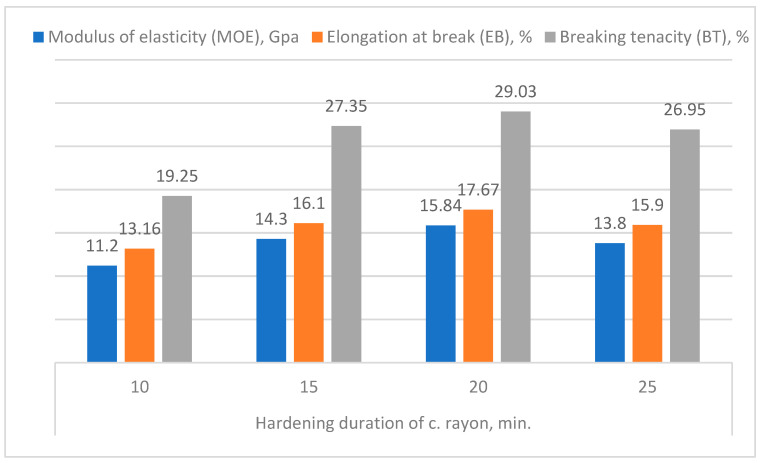
Effect of hardening period of the c. rayon on its mechanical properties, namely modulus of elasticity (MOE), elongation at break (EB), and breaking tenacity (BT).

**Figure 13 polymers-16-02431-f013:**
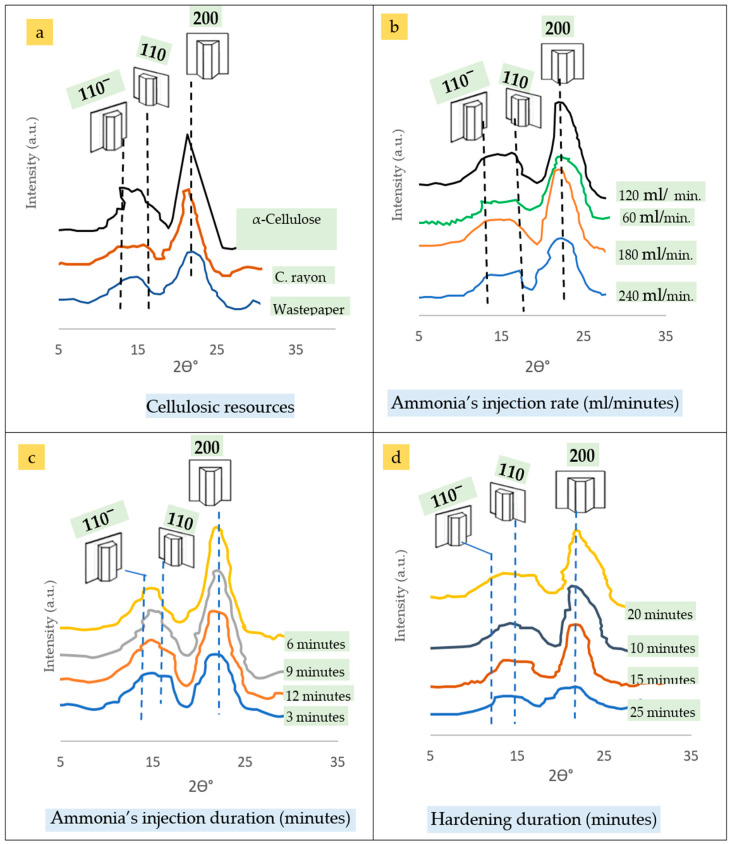
The relationship between crystallinity index (CI, %) and two thetas (degree) as affected by each of the following: (**a**) Cellulosic resources of wastepaper, c. rayon and α-cellulose, and the three fabrication parameters of the c. rayon; (**b**) Ammonia’s injection rate (mL/minutes); (**c**) Ammonia’s injection duration (minutes); (**d**) Hardening duration (minutes).

**Table 1 polymers-16-02431-t001:** Chemical reagent (CR), formula, concentration, and reagent role in the production of cuprammonium rayon.

CR	Formula	Concentration % (wt/wt)	The Reagent Role
Copper sulfate	CuSO_4_	5	Produce copper hydroxide [Cu (OH)_2_].
Sodium hydroxide	NaOH	1	
Copper hydroxide	[Cu (OH)_2_]	80	It is ammoniated to produce the cuoxam solution.
Ammonia gas	NH_3_ (gas)	100	Grafting four [NH_3_^−^] groups on copper hydroxide to produce the cuoxam solution.
Cuoxam solution	[{Cu(NH_3_)_4_}(OH)_2_]	100	Dissolving cellulose
Citric acid	C_6_H_8_O_7_	5	Hardening the rayon fibers in the curing bath.

**Table 2 polymers-16-02431-t002:** (**A**) Calculation of the fibrous properties of the c. rayon synthesized from the wastepaper. (**B**) Calculation of the mechanical properties of the c. rayon synthesized from the wastepaper. (**C**) Calculation of the physical properties of the c. rayon synthesized from the wastepaper. (**D**). Calculation of crystallinity index (CI), mass loss (ML), heat change (HC), glass transition temperature (Tg) of the cuprammonium rayon (c. rayon) as well as its cellulosic precursors (crude and purified wastepaper). (**E**). Calculation of the chemical properties of the c. rayon synthesized from the wastepaper.

(**A**)		
Expression	Equation	Unit
Staple length	Recording by a ruler	mm
Linear density (LD)	LD = 1000 W/L	Tex
W: the air-dry weight of a known length		g
L: the known length of the c. rayon fiber		mm
Fiber diameter	Estimating from SEM images	µm
(**B**)		
Expression	Equation	Unit
Tensile strength (σ_t_)	σ_t_ = F_f_/A	MPa
F_f_: Force at failure		N
Cross-section area (A)	A = π × r^2^ = π (D/2)^2^	(mm)^2^
Fiber diameter (D)	D =2 r	mm
The ratio of circumference to diameter of a circle (π)	π = 22/7	
Modulus of elasticity (MOE)	MOE = δ/ε	GPa
δ: Tensile stress		Pa
Tensile Strain (ɛ)	ɛ = [∆L/L_o_] = [(L_f_ − L_o_)/L_o_]	–
Elongation of the gage length (∆l)	∆L = L_f_ − L_o_	m
L_f_: The final fiber length at failure		mm
L_o_: The initial fiber length at failure		mm
Corrected (MOE_c_)	MOE_c_ = MOE∗/[1 − {C× (E∗ × A/L)}]	GPa
E: the apparent stiffness calculated from the stress–strain curve		
L: gauge length		
Percentage of elongation at failure (E_a_F)	E_a_F = ∆L_f_ = [(L_f_ − L_o_)/L_o_] × 100	%
Breaking tenacity (BT)	BT = [(B/W) × 2.54 × 10^–5^]	g_f_/Tex
B: Filament breaking load		b_f_
W: Filament weight (W)		g
(**C**)		
Expression	Equation/measuring device	Unit
Cellulose yield of the wastepaper (αCY).	αCY = (W_1_/W_2_) × 100	%
W_1_: Oven dry weight of the ∞–cellulose.		g
W_2_: Oven dry weight of the wastepaper.		g
Rayon yield (RY).	RY = (W_3_/W_4_) × 100	%
W_3_: Oven dry weight of the rayon fibers.		g
W_4_: Oven dry weight of the cellulose isolated from the wastepaper.		g
Apparent density (AD).	AD = {D/(D − S)} × ꝭ_medium_	g.cm^–3^
ꝭ_medium_: density of displacement medium.		g.cm^–3^
D: Oven dry weight of the c. rayon fiber.		g
S: Weight of the suspended fiber in water.		g
Moisture Content (MC).	MC = [(W_5_ − W_6_)/(W_5_)] × 100	%
W_5_: Air dry weight of the c. rayon fiber.		g
W_6_: Oven dry weight of the c. rayon fiber.		g
The Moisture Regain (MR).	MR = [(W_7_ − W_8_)/W_8_] × 100	%
W_7_: Water–saturated weight of the c. rayon fiber.		g
W_8_: Oven dry weight of the c. rayon fiber.		g
Volumetric Shrinkage (VS).	VS = [(1 − (L_2_/L_1_)] × 100	%
L_1_: length of the oven-dried C. rayon fiber.		mm
L_2_: length of the swelled c. rayon fiber obtained by immersing in boiling water for 30 s.		mm
(**D**)		
Expression	Equation	Unit
Crystallinity index (CI)	CI = [(Dcr_1_ + Dcr_2_)/Dt] × 100	%
The total peak area (Dt)	Dt = [Ʃ(X_2_ − X_1_)(I_1_ + I_2_)/2]	mm^2^
X_1_: The 1st X-coordinate of the trapezoid constituting a peak area (width).		2θ°
X_2_: The 2nd X-coordinate of the trapezoid constituting a peak width (width).		2θ°
I_1_: The lower Y-coordinate of the trapezoid constituting a peak (intensity).		a.u.
I_2_: The upper Y-coordinate of the trapezoid constituting a peak (intensity).		a.u.
Cristallite size (CS),	CS = Kλ/β_1/2_Cos θ	nm
K: The correction factor is usually taken to be 0.91 (0.1542 nm).		**-**
λ: The radiation wavelength of X-rays incident on the crystal		**-**
β_1/2_: The corrected angular full width at FWHM.		
FWHM: The full width at half maximum of an XRD-peak.		
θ°: The diffraction (Bragg) angle corresponding to the 200 plane.		
Lattice spacing (LS)	LS = nλ/2sin θ°	nm
n: An ordinal number taking a value of “1” for diffractograms having the strongest intensity.		-
Mass loss (ML)	ML % = [{(W1 − W2)/W1}]× 100]	%
W1: Initial rayon weight		g
W2: Final rayon weight after heating		g
W_ct_ = Weight of the purified cellulosic fibrous at a certain moisture content heated from 0–500 °C.		g
Heat change (HC)	using a Seiko & Star 6300 analyzer’s software.	μVs/mg
(**E**)		
Expression	Equation	Unit
Intrinsic viscosity of the α–cellulose dissolved in cuoxam reagent (ղ)	ղ = Km × MW× α	
Km = 8.5 × 10^−3^		mL/g
MW: Molecular weight		g/mole
α = 0.81		
Degree of polymerization (DP)	DP = MW_1_/MW_2_	–
MW_1_: Total molecular weight of the rayon fiber.		g/mole
MW_2_: Molecular weight of the glucopyranose monomer.		g/mole

**Table 3 polymers-16-02431-t003:** Mean values ^1^ of the different properties of the cuprammonium rayon (c. rayon) synthesized from the wastepaper.

	Property		C. rayon	Standard Limits	
				Value	References
Fibrous	Staple length, mm		44 ± 4.8	39	[66]
	Linear density, Tex		235 ± 4.31	94–244	[23,70]
	Fiber diameter, µm		19.4 ± 1.49	54	[23]
Mechanical	Tensile strength, MPa		218.3 ± 3.37	360	[99,100,101]
	Modulus of elasticity, GPa		14.3 ± 0.28	0.5 ± 4–11 ± 4.3	
	Elongation at break, %		16.1 ± 0.33	5.3 ± 1.9–14 ± 3.8	
	Breaking tenacity cN/Tex		27.53 ±0.41	35 ± 3	
Physical	Yield α–cellulose, %		90.3 ± 0.81	84.76	[2]
	Rayon yield, %		92.25 ± 1.17	72.47–88.27	[16]
	Apparent density, g.cm^–3^		1.54 ± 0.16	1.53	[97]
	Moisture content, %		8.6 ± 0.98	10.2–12.75	[16,66]
	Moisture regain, %		7.8 ± 0.76	10.57–14	[16,20,91]
	Volumetric shrinkage, %		1.8 ± 0.24	1.5	[96]
	Crystallinity index (CI), %		61.304	71.62	[110]
	Maximum mass loss (25 °C–500 °C), %		56.77		[47,90]
	Heat change, μVs/mg	Endotherm	−884	−784.49	[11,47,90]
		Exotherm	+1247	879.29	
		Net energy	363	94.8	
	Glass transition temperature (Tg), °C		220°		[86]
Chemical	Molecular weight, g/mol		64,800 ± 471	90,000–110,000	[97]
	Degree of polymerization, Dalton		400 ± 4.9	285–603.4	[98]

^1^ Each value is an average of 3 samples followed by the standard deviation.

## Data Availability

The original contributions presented in the study are included in the article/Appendix A, further inquiries can be directed to the author.

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
