# Peer review of "Wastepaper-Based Cuprammonium Rayon Regenerated Using Novel Gaseous–Ammoniation Injection Process"

_polymers, 2024, doi:10.3390/polym16172431_

Round 1

Reviewer 1 Report

Comments and Suggestions for Authors

This paper introduces a novel process called Gaseous-Ammoniation Injection (GAI) for synthesizing cuprammonium rayon from wastepaper. This work is significant in the global textile industry as the process is more environmentally friendly and the raw material is recycled. The paper provides a comprehensive description of the production process and mathematical description of characterizations. The characterization of the resulting cuprammonium rayon fibers demonstrate that C. Ryan Fibers can achieve with the novel process in comparable qualities. The mechanical properties measured at different process parameters reveal the optimal GAI conditions, including the injection rate of ammonia gas, the injection duration and hardening period. Overall, this paper is a substantial contribution and is suitable for publication.

Here are some minor changes suggested:

L297: what LS means here? Is it a typo? If not, please give whole name that LS indicates here and explain it.

Table 2C: please highlight fifth row.

Figure 5: please unhighlight the blue color and label which figure is d.

Line 478: delete extra space before “melting point”.

Line 590: Correct “Table 12” to “Figure 12”.

Author Response

Open Review

Quality of English Language

( ) I am not qualified to assess the quality of English in this paper
( ) English very difficult to understand/incomprehensible
( ) Extensive editing of English language required
( ) Moderate editing of English language required
( ) Minor editing of English language required
(x) English language fine. No issues detected

Yes

Can be improved

Must be improved

Not applicable

Does the introduction provide sufficient background and include all relevant references?

(x)

( )

( )

( )

Is the research design appropriate?

(x)

( )

( )

( )

Are the methods adequately described?

(x)

( )

( )

( )

Are the results clearly presented?

(x)

( )

( )

( )

Are the conclusions supported by the results?

(x)

( )

( )

( )

Comments and Suggestions for Authors

The general opinion of the reviewer: This paper introduces a novel process called Gaseous-Ammoniation Injection (GAI) for synthesizing cuprammonium rayon from wastepaper. This work is significant in the global textile industry as the process is more environmentally friendly and the raw material is recycled. The paper provides a comprehensive description of the production process and mathematical description of characterizations. The characterization of the resulting cuprammonium rayon fibers demonstrates that C. Ryan Fibers can achieve with the novel process in comparable qualities. The mechanical properties measured at different process parameters reveal the optimal GAI conditions, including the injection rate of ammonia gas, the injection duration and hardening period. Overall, this paper is a substantial contribution and is suitable for publication.

1

Reviewer’s remark

L297: what LS means here? Is it a typo? If not, please give whole name that LS indicates here and explain it.

My comment

LS is not typo but abbreviation of staple length of c. rayon. Under the title of “2.5.1. Fibrous Properties”, the next paragraph refers to the meaning of the “SL”, “LD” and “FD” (Page 11, lines 288-290).

“Three of the essential fibrous properties, namely staple length (SL), linear density (LD), and fiber diameter (FD) were investigated, and their evaluation was conducted as illustrated in Table 2A”. In addition, the same meaning of the “SL”, “LD” and “FD” was incorporated within Table 2A.

However, based on your respective comment, I deleted these abbreviations, and I presented the whole names.

2

Reviewer’s remark

Table 2C: please highlight fifth row.

My comment

Now, it is highlighted.

3

Reviewer’s remark

Figure 5: please unhighlight the blue color and label which figure is d.

My comment

Done

4

Reviewer’s remark

Line 478: delete extra space before “melting point”.

My comment

Done

5

Reviewer’s remark

Line 590: Correct “Table 12” to “Figure 12”.

My comment

Done

Submission Date

15 June 2024

Date of this review

29 Jun 2024 05:27:52

Reviewer 2 Report

Comments and Suggestions for Authors

The authors presents a new invention of using Gaseous–Ammoniation Injection (GAI) Process to generate Cuprammonium Rayon. The overall results are soundness. However, there are some error which makes reader confuse.

1. Figure 3 number label and description do not mach. Same as Figure 4. Please check all Figure number label and description matches so that reader don't get confused.

Besides this, I have no comment on the article.

Figure 3 number label and description. Same as Figure 4

Author Response

Open Review

Quality of English Language

( ) I am not qualified to assess the quality of English in this paper
( ) English very difficult to understand/incomprehensible
( ) Extensive editing of English language required
( ) Moderate editing of English language required
( ) Minor editing of English language required
(x) English language fine. No issues detected

Yes

Can be improved

Must be improved

Not applicable

Does the introduction provide sufficient background and include all relevant references?

(x)

( )

( )

( )

Is the research design appropriate?

(x)

( )

( )

( )

Are the methods adequately described?

(x)

( )

( )

( )

Are the results clearly presented?

(x)

( )

( )

( )

Are the conclusions supported by the results?

(x)

( )

( )

( )

Comments and Suggestions for Authors

The general opinion of the reviewer:  The author presents a new invention of using Gaseous–Ammoniation Injection (GAI) Process to generate Cuprammonium Rayon. The overall results are soundness. However, there are some errors which confuse reader confuse.  Besides this, I have no comment on the article.

1

Reviewer’s remark

Figure 3. number label and description do not match. Please check all Figure number label and description matches so that reader don't get confused.

My comment

All mistakes were corrected as presented at page 8 of the manuscript and as follow:

Figure 3. Production of the rayon fibers: a) CuSO4, b) NaOH, c) Cu (OH)2, d) The dis-carded NaSO4, e) the novel gaseous ammoniation process: e1) the gaseous vessel, e2) flowmeter, f) α–cellulose that will dissolve in cuoxam solution, g) the dark blue color of the recently–prepared cuoxam solution (tetra–ammine copper hydroxide): g1) chemical structure of the cuoxam solution, g2) the atoms’ types constituting the cuoxam sol-vent, h–j) optical images of the hardened rayon fibers: h) curing of the rayon fibers within the hardening bath, whereby the blue threads are recent–extruded ones within the hardening bath containing citric acid, while the white ones are recently–cured that lost their copper ions, i) a close up of recently–cured rayon fibers, j) air–drying the fab-ric after neutralization and washing, and k) oven-dried rayon fibers. 

2

Reviewer’s remark

Figure 4. number label and description do not match.  Please check all Figure number label and description matches so that reader don't get confused.

My comment

All mistakes were corrected as presented at page 8 of the manuscript and as follow:

Figure 4. The manufacture of the cuprammonium textile: A) woven fabric: a1-a7) the manual weaving of staple fibers synthesized from the wastepaper: a1) staple fibers, a2) primitive carding machine, a3) carding brush to get ordered and aligned fibers, a4) spinning the ordered fibers into using a hand–spindle, a5) threads, a6) textile weaving machine, a7) a typical textile, a8) stretch pattern of plain weave 1/1, a9) stretch pattern of panama weave 2/2, and B) non-woven fabrics: b1) randomly arranged-fibers, b2) two–dimensional textile sheet, b3) a close up of a sheet showing a loosen permeable spot (b4–b7)  and a welded spot (b5,b6) arose upon thermal compression of the randomly arranged-fibers.  

In addition, the next paragraph (pages 10-11, lines 269-286) was modified:

Production of Non–Woven Fabric

Nonwoven fabrics consist of randomly arranged sheets of fibers that are bonded together through adhesive bonding, entanglement, or sewing. Cords and ropes utilize two–dimensional (2D) constructions. Three–dimensional (3D) fabrics are utilized as composite preforms, either in the configuration of molded sheets or substantial constructions. Additionally, they find application in other areas such as knitted clothing and conveyor belts [51]. The nonwoven textile was fabricated in this investigation by extrusion of the rayon fibers within the curing or hardening solution on a perforated plate in a duplicate layer whereby any sublayer was perpendicular to the other one (Figure 4). No adhesives were used for binding the fibers into the 3D structure of the non–woven fabric. Welding the loosen permeable spot shown in Figure 4b1 into a sheet (Figure 4b2) was achieved using a thermal spot–welding machine using a red copper–protrusions. Accordingly, loosen permeable spots (Figure 4b4–4b7) and welded spots (Figure 4b5,4b6) were obtained offering good permeability and reinforcing the non-woven fabric, respectively.

It is worth for mentioning that binding forces responsible for attracting the cellulosic chains within/between fibers (intra and inter, respectively) were strongly believed to be hydrogen bonding and Van der Waals forces (Figure 5). These forces have arisen between the negative hydroxyl groups (OH) and H+ belonging to the cellulose macromolecule.

Submission Date

15 June 2024

Date of this review

21 Jun 2024 04:23:44

Reviewer 3 Report

Comments and Suggestions for Authors

The Sherif S. Hindi manuscript is devoted to the production of hydrated cellulose fibers from paper waste. In my opinion, the topic is interesting and relevant. From my own experience, I can say that there is increased activity in this direction by a number of scientific groups in a number of countries. The volume of the manuscript and the amount of literature used is more suitable for a review work. Perhaps the text can be simplified and shortened?!
I would like to immediately suggest removing the abbreviation (GAI) from the title of the manuscript. It is better to introduce it in the abstract.

What do the authors mean by the term "Hardening solution"?

The presented literature review does not lead to a specific goal of the work. Therefore, it is necessary to formulate a clear goal before considering materials and methods.

In addition to Calcium Carbonate, other impurities, such as metals, may be present in cellulose raw materials. How will they affect the stability of the process? What are the permissible concentrations?
In Figure 2, the figure caption is not entirely correct. It is necessary to check and correct.

25. The authors must decipher the abbreviation - TS.
56-59. I don't see NMMO in the list of solvents used. This solvent is widely used and must be added. Its use is described in this work - Sevastyanova, J.V. et al. Modern Technology for the Production of Hydrated Cellulose Fibers. Fibers Polym 25, 913–921 (2024). https://doi.org/10.1007/s12221-024-00485-9

In Figure 3, the molded extrudates are at least 100 µm thick, while the micrograph shows that the diameter is on the order of 20 µm?

The methodological part can be shortened!

In Table 3 I don't understand some of the values, for example in the line MP (°C)? Above, the authors write that cellulose is not a thermoplastic.
There is no dimension in the figure (X axis).
Why do my strength characteristics have low values?
The data presented in the diagram, line 560, has no meaning?!
3.2.3.1. Melting Point (MP) - this part is not clear to me?!

There are a lot of typos and blots in the work, so in 107 a period is missing, in 420 the title needs to be highlighted, etc.

The bibliography consists of 103 titles, which confirms the great work done. However, this list is not entirely accurate and needs to be corrected.

In general, the work, in my opinion, requires deep processing. Now the manuscript looks like a student's thesis. Also, not obvious novelty allows me to suggest that the authors submit this work to the journal Materials (MDPI), which is dedicated specifically to materials.

Author Response

Open Review

Quality of English Language

(x) I am not qualified to assess the quality of English in this paper
( ) English very difficult to understand/incomprehensible
( ) Extensive editing of English language required
( ) Moderate editing of English language required
( ) Minor editing of English language required
( ) English language fine. No issues detected

Yes

Can be improved

Must be improved

Not applicable

Does the introduction provide sufficient background and include all relevant references?

( )

( )

(x)

( )

Is the research design appropriate?

( )

( )

(x)

( )

Are the methods adequately described?

( )

( )

(x)

( )

Are the results clearly presented?

( )

( )

(x)

( )

Are the conclusions supported by the results?

( )

( )

(x)

( )

Comments and Suggestions for Authors

The general opinion of the reviewer: Thismanuscript is devoted to the production of hydrated cellulose fibers from paper waste. In my opinion, the topic is interesting and relevant.

In general, the work, in my opinion, requires deep processing

1

Reviewer’s remark

Perhaps the text can be simplified and shortened

My comment

The new title is:

Wastepaper based Cuprammonium Rayon Using Novel Gaseous–Ammoniation Injection Process

2

Reviewer’s remark

I would like to immediately suggest removing the abbreviation (GAI) from the title of the manuscript. It is better to introduce it in the abstract.

My comment

The abbreviation (GAI) was removed from the manuscript’s title and it was introduced to the abstract.

3

Reviewer’s remark

What do the authors mean by the term "Hardening solution"?

My comment

The following paragraph was modified, and added in the introduction section (page 3, lines 101-109).

“Rayon is produced by dissolving cellulose in a solution called Cuoxam, which is a deep blue solution containing tetra–ammine cupric hydroxide, also known as Cuoxam (Schweitzer) reagent. The latter is achieved by combining a copper sulphate solution with NaOH to form cupric hydroxide, which is then dissolved in NH4OH solution and extruded through a spinneret into a hardening bath known as coagulation bath hardening process typically involves immersing the fibers in a chemical solution that contains a coagulant, such as sulfuric acid or aluminum sulfate or others. The coagulant reacts with the fiber's surface, causing the molecules to aggregate and form a more rigid structure. Rayon is produced by dissolving cellulose in a solution called Cuoxam, which is a deep blue solution containing tetra–ammine cupric hydroxide, also known as Cuoxam (Schweitzer) reagent. The latter is achieved by combining a copper sulphate solution with NaOH to form cupric hydroxide, which is then dissolved in NH4OH solution and extruded through a spinneret into a hardening bath known as coagulation bath hardening process typically involves immersing the fibers in a chemical solution that contains a coagulant, such as sulfuric acid or aluminum sulfate or others. The coagulant reacts with the fiber's surface, causing the molecules to aggregate and form a more rigid structure”.

4

Reviewer’s remark

In addition to Calcium Carbonate, other impurities, such as metals, may be present in cellulose raw materials. How will they affect the stability of the process? What are the permissible concentrations?

My comment

I added the following paragraph into page 25, lines 723-726 in the future prospective section:

“Regarding calcium carbonate and other impurities, such as metals, may be present in cellulose raw materials, studying how may they affect the stability of the cuprammonium rayon and determining, accurately the permissible concentration of each pollu-tant is very important for future investigations.”

In addition, the next paragraph was introduced to the introduction section (page 3, lines 133-137):

“While there is no single, universally accepted concentration limit for calcium carbonate in rayon, generally accepted limits range from 0.1% to 1.0% by weight of the fiber, depending on the application, quality requirements, and regulatory standards. However, I added this paragraph to the Introduction section, and directed it to the fu-ture prospective due to the importance of your remark.”

5

Reviewer’s remark

In Figure 2, the figure caption is not entirely correct. It is necessary to check and correct.

My comment

Done in page 5. I fixed the arrows’ positions in Figure 2.

6

Reviewer’s remark

The authors must decipher the abbreviation - TS.

My comment

Ok, it was done all over the manuscript including Table 3 in page 17:

Table 3. Mean values 1,2 of the physical properties of the cuprammonium rayon (c. rayon) synthesized from the wastepaper, namely yield of α–cellulose and the rayon properties of melting point, rayon yield, apparent density, moisture content, moisture regain, volumetric shrinkage, and chemical properties, namely molecular weight, degree of polymerization and the fibrous properties, namely staple length, linear density, and fiber diameter, and the mechanical properties (MP), namely tensile strength, elongation at break, modulus of elasticity, and breaking tenacity.

Property

C. rayon

Standard limits

Value

References

Melting point, °C

181° ± 1.43°

149

[70]

Yield α–cellulose, %

90.3 ± 0.81

84.76

[2]

Rayon yield, %

92.25 ± 1.17

72.47–88.27

[16]

Apparent density, g.cm–3

1.54 ± 0.16

1.53

[91]

Moisture content, %

8.6 ± 0.98

10.2–12.75

[66],[16]

Moisture regain, %

7.8 ± 0.76

10.57–14

[16,20,92]

Volumetric shrinkage, %

1.8 ± 0.24

1.5

[90]

Molecular weight, g/mol

64,800 ± 471

90,000– 110,000

[91]

Degree of polymerization, Dalton

400 ± 4.9

285–603.4

[92]

Staple length, mm

44 ± 4.8

39

[66]

Linear density, Tex

235 ± 4.31

94–244

[23,70]

Fiber diameter, µm

19.4 ± 1.49

54

[23]

Tensile strength, MPa

218.3 ± 3.37

360

[93]

Modulus of elasticity, GPa

14.3 ± 0.28

0.5 ± 4–11 ± 4.3

[94]

Elongation at break, %

16.1 ± 0.33

5.3 ± 1.9–14 ± 3.8

Breaking tenacity cN/Tex

27.53 ± 0.41

35 ± 3

[95]

7

Reviewer’s remark

56-59. I don't see NMMO in the list of solvents used. This solvent is widely used and must be added. Its use is described in this work - Sevastyanova, J.V. et al. Modern Technology for the Production of Hydrated Cellulose Fibers. Fibers Polym 25, 913–921 (2024). https://doi.org/10.1007/s12221-024-00485-9

My comment

Yes Prof. I presented some ordinary solvents of cellulose not all the used. According to your valuable comment, I referred shortly to this solvent. It is famous, but I focused here to the Cuoxam solvent due to cheapness, easiness and environmental concerns

8

Reviewer’s remark

In Figure 3, the molded extrudates are at least 100 µm thick, while the micrograph shows that the diameter is on the order of 20 µm?

My comment

When I revised the dimensions in Figure 3, I assure that they are correct, since the length of the ruler present in the Figure 4k is 100 µm, and the fiber width is 19.4µm, about fifth the ruler. Thank you, Sir, for your great effort.

9

Reviewer’s remark

The methodological part can be shortened!

My comment

Ok sir, the methodology was abstracted as you advised.

10,11

Reviewer’s remark

·        In Table 3 I don't understand some of the values, for example in the line MP (°C)? Above, the authors write that cellulose is not a thermoplastic.

·        3.2.3.1. Melting Point (MP) - this part is not clear to me?!

My comment

In page 14, lines 394-401, there is clear illustration of the melting point of the c.rayon:

“Melting point (MP) of the c. rayon is a useful indicator of its identity, purity, charac-teristics and behavior [78]. For high pure solid substance, their melting point was found to be lied within a very narrow range, about 1 °C, or less [79]. Concerning de-termining the MP, the oven–dried specimens of rayon fibers were grinded into fine powder (100 mesh). Then, the melting point determination was done using Standard Steel Mild Steel Melting Point Apparatus (MP01). Calibration of a melting point meas-uring instrument was carried out by observing the melting behavior of a benzoic acid powder as a reference standard with a melting point of 122.383 °C [78].”

In addition, the following paragraph is present already in the text (In page 23, lines 636-641)

The melting point of the rayon fibers was found to be about 181 °C that is in accordance with that for the standard value (above 150°C). In addition, the melting point average was greater than that measured by Shamsuddin et al. [70] as shown in Table 3. Accordingly, the melting point of the invented rayon is satisfactory, since higher MP reflects thermal resistance of the fibers. The higher the melting point, the higher the quality of the resulting fabrics.

12

Reviewer’s remark

There is no dimension in the figure (X axis).

My comment

For Figures 8, 10, 12, I presented collective histograms for the three properties, namely modulus of elasticity, elongation at break, and breaking tenacity where each of them is a Y-variable.

13

Reviewer’s remark

Why do my strength characteristics have low values?

My comment

I added the following the next paragraph in the final of the manuscript to cover your comment:

“Although the mechanical properties’ values were satisfactory, an enhancement’ s target in the future prospective must be considered. “

14

Reviewer’s remark

The data presented in the diagram, line 560, has no meaning?!

My comment

“Presenting the influence of the mechanical properties of c. rayon by the most important processing parameters is clear in Figures 7–12.

Concerning the ammonia’s injection rate upon synthesis cuoxam reagent, it is obvious from Figure 7 that the tensile strength was increased significantly from 179.2 MPa to 218. 3 MPa when the injection rate was raised from 60 to 120 ml/minute. Furthermore, there was no significant change in the TS trend above the injection rate of 120 ml/min.” [page 20, lines 548-553]

Moreover, the next paragraph “It is worth mentioning that the same trend recorded for the tensile strength was re-peated for the other tensile properties studied (modulus of elasticity, elongation at break, breaking tenacity). For an excess of illustration, the injection rate of 120 ml/min. revealed to the highest mean values of the MOE, EB, and BT (14.3 GPa, 16.1 %, and 27.35 %, respectively) as indicated in Figure 8.” [page 20, lines 556-559]:

From the above-mentioned two paragraphs, it is clear that…..

15

Reviewer’s remarks

·        There are a lot of typos and blots in the work

·        In 107 a period is missing

·        In 420 the title needs to be highlighted, etc.

My comment

The manuscript was corrected as clear in the case of the colored-highlighted words and/or statements all over its text.

For Figure 5

16

Reviewer’s remark

The bibliography consists of 103 titles, which confirms the great work done. However, this list is not entirely accurate and needs to be corrected.

My comment

Revised accurately Sir as clear in the 1st round-revised version.

Submission Date

15 June 2024

Date of this review

22 Jun 2024 20:19:21

Round 2

Reviewer 3 Report

Comments and Suggestions for Authors

Answers have been received to the questions posed earlier, but some of them do not completely satisfy me. For example, I do not understand the given melting point values (Table 3). The authors claim that cellulose fibers are thermoplastics? But it is known that cellulose has a melting point below its destruction.
Figure 6. The dimension for Stress (ε) is not given.
The bibliography is incorrect and needs to be corrected!

Author Response

Comments and Suggestions for Authors and my reply to the respective reviewer

Answers have been received to the questions posed earlier, but some of them do not completely satisfy me.

Remark 1.  I do not understand the given melting point values (Table 3). The authors claim that cellulose fibers are thermoplastics. But it is known that cellulose has a melting point below its destruction.

My reply 1.

The next paragraph was introduced (page 17, lines: 471-476) illustrating the melting phenomenon:

Deeply speculating Figure 5d revealed that some cellulose vapors were emitted from the hot surface of the sample under vacuum. These ideal conditions for melting of c. rayon were dominated at the samples’ chamber of the SEM device. Accordingly, these scarce images were obtained coincidently upon higher focusing imaging of the fibrous c. rayon’s structure since increasing focusing requires rising the dominating voltages that is responsible for elevating temperature around the sample upon imaging. It is worth noting that the bar scale is not related to the actual dimensions of the cellulose’s vapor which is in the gaseous state.

The next paragraph was introduced under the future prospective section illustrating the melting phenomenon (page 25, lines: 732-733): In addition, thermal behavior of rayon, especially melting trend as well as its affecting by endo- and exothermic arisen upon differential thermal degradation as a trial to enhancing the thermal characteristics of the rayon fibers.

Furthermore, the red- highlighted statement in the next paragraph was added:

“Melting point (MP) of the c. rayon is a useful indicator of its identity, purity, characteristics and behavior [78]. For high pure solid substance, their melting point was found to be lied within a very narrow range, about 1 °C, or less [79]. Concerning determining the MP, the oven–dried specimens of rayon fibers were grinded into fine powder (100 mesh). Then, the melting point determination was done using Standard Steel Mild Steel Melting Point Apparatus (MP01). Calibration of a melting point measuring instrument was carried out by observing the melting behavior of a benzoic acid powder as a reference standard with a melting point of 122.383 °C [78] (page 14).”

In addition, the next paragraph was added to the text based on your valuable remark (page 24, line s 648-660)

For shedding more light over the relation of melting point of the rayon and its crystallographic behavior illustrating It was stated by Shamsuddin et al. [70] that two distinct melting peaks with varying strengths were observed in the differential scanning calorimetry curves of their fabricated rayon: one at approximately 150 °C and the other at 162 °C. These melting peaks were ascribed to either (i) the presence of two lamella structures in their samples or (ii) the first melting and subsequent re-crystallization of composite’s crystals, which may have an impact on how these crystals reorganize during the second heating. However, during cooling, for both rayon fiber-reinforced (NFC-reinforced) and unreinforced composite samples, a single re-crystallization peak was seen at roughly 79 °C. Additionally, it is clear that the upper melting peaks were more prominent, indicating that the reinforced composite samples contain is primarily composed of stable crystals with a tiny amount of unstable crystals. As a result, the two melting peaks might be connected to these crystals' reorganization during the second heating. In addition, all produced composites had a melting temperature shift of roughly 8 °C higher than that of the unreinforced composite samples.

Additional reply from me belonging my satisfied results is clear from the next paragraph that was present already in the text (page: 23, lines: 643-648) and referenced by Shamsuddin et al [70]:

“The melting point of the rayon fibers was found to be about 181 °C, that is in accordance with the standard value (above 150°C). In addition, the melting point average was greater than that measured by Shamsuddin et al. [70] as shown in Table 3. Accordingly, the melting point of the invented rayon is satisfactory, since higher melting point reflects thermal resistance of the fibers. The higher the melting point, the higher the quality of the resulting fabrics. This paragraph illustrates the higher melting point of my c. rayon compared than that synthesized by Shamsuddin et al [70] as clear in Table 3 (page 18).

 Remark 2. 
Figure 6. The dimension for Stress (ε) is not given.

My reply 2.

I am sorry, there was a mistake in Figure 6 concerning the “ε” for the X-coordinate, it is strain not stress. The latest variable is found already on the Y-coordinate with its unit (MPa). I corrected the word “stress” to be “Strain (ε), %” for the X-coordinate. It is worth mentioning that the strain has a percentage unit since it is arising from dividing length by length multiplied by 100.

Remark 3. 
The bibliography is incorrect and needs to be corrected!

My reply 3.

I found one mistake in my call number, and I corrected it as shown and blue-highlighted bellow:

1 Department of Agriculture, Faculty of Environmental Sciences, King Abdulaziz University (KAU), P.O. Box 80208, Jeddah–21589, Saudi Arabia; shindi@kau.edu.sa; 009566760086.

Other reply from me based on the reviewers’ notices:

For Table 2C (page 14, lines 392-393), see the word file.

The reference section (pages 26-30) was revised, the duplicated similar numbers in the same raw were corrected.

Under the next title “3.2.3.6. The Moisture Regain (MR)” there was two small paragraphs that merged together to constitute the following, green-highlighted paragraph:

“The MR values of cotton and regenerated cellulose are about 7.8 % (Table 3). Due to its cellulosic nature, viscose has more amorphous regions giving more voids in its structure [102]. This nanostructure of the rayon fibers has some moisture management properties among all viscose blends. It was found to have moisture absorption ability, but it wicks less as compared to natural fibers such as cotton fibers [66,102].”

3.2.3.7. The Volumetric Shrinkage (VS)

In addition, the following paragraph was confounded from three smaller ones “The VS’ mean value of the rayon fibers was determined to be about 1.8 % that lies within the accepted region. This value is higher to some extent than that found by Nawaz et al. [90] as presented in Table 3. Since volumetric shrinkage is the dimensional change resulting in a decrease in the length or width of a rayon fiber specimen due to losing some of its moisture content, their fabrics’ dimensions are expected to decrease slightly when exposed to heat or evaporation conditions. However, this VS was small enough to be undistorted for the final product fabrics of the rayon fibers. Since rayon fibers are composed mainly from cellulose that is well known to be hydrophilic in its nature, it can allow the water to soak into the fiber and swell as well as losing moisture and is shrunk. On the other hand, hydrophobic fibers such as cellulose triacetate will exhibit very little shrinkage [103].”

Concerning the “Results and Discussion”

……”Cuoxam solution”….. was converted to be ……”cuoxam solution”….. within a statement.

Under the title of 3.2.1. Fibrous Properties, the next subtitles were discarded with preserving their paragraphs:

3.2.1.1. The Staple Length (SL)

3.2.1.2. The Linear Density (LD)

3.2.1.3. The Fiber Diameter (FD)

Under the title of 3.2.2. Mechanical Properties of the Cuprammonium Rayon, the next subtitles were discarded with preserving their paragraphs:

3.2.2.1. Tensile Strength (TS)

3.2.2.2. Modulus of Elasticity (MOE)

3.2.2.3. Elongation at Break (EB)

3.2.2.4. Breaking Tenacity (BT)

Page 19: the next paragraph was confounded from three smaller ones:

The tensile Strength’ s of the rayon 218.3 MPa as indicated from Table 3. The lower strength of the rayon fibers obtained in the present study comparing to that for regenerated cellulose (360 MPa) fabricated by Dirgar [93] can be attributed to the lower crystallinity and higher amorphous regions in the rayon structure and vice versa for Tencel material [97–99]. Changes in tensile strength and elongation with wetting may depend mainly on the number of the molecular chain ends in the amorphous region [59].

The subtitles of The Tensile Strength (TS), Modulus of Elasticity (MOE), Elongation at Break (EB), 3.2.2.4. Breaking Tenacity (BT) were discarded (pages 19, 20).

Under the title “3.2.3. Physical Properties”:

the next subtitles were discarded with preserving their paragraphs:

3.2.3.1. Melting Point

3.2.3.2. α–cellulose yield (αCY)

3.2.3.3. The Rayon Yield (RY)

3.2.3.4. The apparent density (AD)

3.2.3.5. The Moisture Content (MC)

3.2.3.6. The Moisture Regain (MR)

Moreover, several modifications were made concerning introduction, research design, methods, results’ presentation, and conclusions based on your valuable remarks.

Thanks so much for enhancing my manuscript.

Round 3

Reviewer 3 Report

Comments and Suggestions for Authors

I do not agree with the authors' answers regarding the melting point of cellulose fibers. It is known that the temperature of cellulose destruction is lower than its melting temperature. If it were the other way around, then cellulose fibers would be produced by the melt process. Structural and morphological data are required to evaluate the resulting fibers. It is also not entirely clear to me how stable the spinning will be due to the presence of impurities in the feedstock.

Author Response

Comments and Suggestions for Authors:

Reviewer’ s Comment 1:

The respective reviewer does not agree with the author' answers regarding the melting point of cellulose fibers. It is known that the temperature of cellulose destruction is lower than its melting temperature. If it were the other way around, then cellulose fibers would be produced by the melt process.

Author’s reply 1:

Dear respective reviewer, at first, I suggest changing the expression ”melting point” into synonymous of softening point. I think that the latter synonymous is closest to imagine of ordinary reader although they are adopted in their scientific thermal concept. In addition, I focused on the “glass transition temperature (Tg) property” rather than the “melting point expression”. I think that the Tg is more suitable for big molecules having degree of polymerization like cellulose, since is an important parameter affecting the material's mechanical properties, thermal stability, and suitability for various applications.

Accordingly, I made the following modifications:

Section

Page

Line

Materials and Methods

Determination of glass transition temperature (Tg)

The Tg is an important parameter in material science, as it affects the material's mechanical properties, thermal stability, and suitability for various applications.

Results and Discussion

Modified as clear in the revised version of the manuscript.

Newly-added references

For glass transition temperature

81-Szczesniak, L.; Rachocki, A.; Tritt-Goc, J. Glass transition temperature and thermal decomposition of cellulose powder. Cellulose. 2008, 15:445–451.

82-Ciolacu, D.; Popa, V. On the thermal degradation of cellulose allomorphs. Cell. Chem. Technol. 2006. 40, 445–449.

83-Goring, D.A.I. Thermal softening of lignin, hemicellu lose and cellulose. Pulp Paper Mag. Can. 1963, 64, T517–T527.

84-Hancock, B.C.; Zografi, G. The relationship between the glass phase transition temperature and the water content of amorphous pharmaceutical solids. Pharmaceut. Res. 1994, 11, 471–477.

85-Salme´n, N.L Back EL The influence of water on the glass phase transition temperature of cellulose. TAPPI. 1977, 60, 137–140.

86-Wunderlich, B. Thermal analysis of polymeric materials. Springer, Berlin, Heidelberg, New York. 2005.

87-Salmén, N.L.; Back. E. The influence of water on the glass transition temperature of cellulose. In Fibre-Water Interactions in Paper-Making, Trans. of the VIth Fund. Res. Symp. Oxford, 1977, (Fundamental Research Committee, ed.), pp 683–690, FRC, Manchester, 2018.

For thermal analysis

110- Smole, M.; Persin, Z.; Kreze, T.; Stana-Kleinschek, K.; Ribitsch, V.; Neumayer, S. Materials Research Innovations. X-ray study of pre-treated regenerated cellulose fibres. 2003, 7, 275-282.

111-Wulandari, W.T.; Rochliadi, A.; Arcana, I.M. Nanocellulose prepared by acid hydrolysis of isolated cellulose from sugarcane bagasse. IOP Conf. Series. J. Mater. Sci. Eng. 2016, 107, 012045.

112-Terinte, N.; Ibbett, R.; Schuster, K.C. Overview on native cellulose and microcrystalline cellulose I structure studied by X–ray diffraction (WAXD): Comparison between measurement techniques. Lenzinger Berichte. 2011, 89, 118–131.

113-Park, S.; Baker, J.O.; El–Himmell, M.; Parilla, P.A.; Johnson, D.K. Cellulose crystallinity index: Measurement techniques and their impact on interpreting cellulase performance. Biotechnol. for Biofuels. 2010, 3 (10).

114-Kumar, A.; Negi, Y.S.; Choudhary, V.; Bhardwaj, N.K.Characterization of cellulose nanocrystals produced by acid–hydrolysis from sugarcane bagasse as agro–waste. J. Mater. Phys. Chem. 2014, 2, 1–8.

115.         Kargin, V. A., Kozlov, P. V. and Wang, Nai-Ch'ang., Doklady Akad. Nauk. SSSR, 1960, 130 (2), 356-8

116.         Alfthan, E., de Ruvo, A. and Brown, W., Polymer, 1973, 14, 329

117.         Goring, D. A. I., Pulp Pap. Mag. Can., 1963, 64 (12), T-517

118.         Back, E. L. and Didriksson, E. I. E., Svensk Papperstid., 1969, 72 (21), 687

119.         Naimark, N. I. and Fomenko, B. A., Vysokomol. Soyed., 1971, B13, 45

120.         Kaimins, I. F. and Ioelovich, M. Ya., Vysokomol. Soyed., 1973, B15 (10), 764

121.         Kaimins, I. F. and Ioelovich, M.Ya., Khim. Drev., 1974, (2), 10-1

122. Kaelbe, D. H., Physical Chemistry ofAdhesion, (Wiley Interscience, New York, 1971).

123. Sisson, W.A. X-ray studies of crystallite orientation in cellulose fibers. III. Fiber structures from coagulated cellulose. J. Phys. Chem. 1940, 44, 513–529.

124. Ismail, M.Y.; Sirviö, J.A.; Ronkainen, V.P.; Patanen, M.; Karvonen,V.; Liimatainen, H. Delignification of wood fibers using a eutectic carvacrol–methanesulfonic acid mixture analyses of the structure and fractional distribution of lignin, cellulose, and hemicellulose. Cellulose. 2024, 31, 4881–4894.

125. Pancholi, M.J; Khristi, A.; Athira, K.M; Bagchi, D. Comparative analysis of lignocellulose agricultural waste and pre-treatment conditions with FTIR and machine learning modeling. BioEnergy Res. 2023, 16, 123–137.     

Deleted references

“Young, J.C. True melting point determination. Chem. Educator. 2013, 18, 203–208. (its original no. was 78)”.

Reviewer’ s Comment 2:

Structural and morphological data are required to evaluate the resulting fibers.

Author’s reply 2:

You are true Prof., structural properties are those relating force to deformation or stress to strain and are crucial for determining their mechanical performance, durability, and suitability for various applications (Ochshorn, 2010). Accordingly, these properties comprise mechanical properties (Tensile strength, modulus of elasticity, elongation at break, and breaking tenacity), as well as XRD features (crystallinity index, crystallite size and lattice spacing). Already these properties were determined and presented at Figures 6-12 (for the results) and Table (2B for calculations), Based on your valuable advice, XRD output of the c. rayon fiber was added as presented at Figure and Table.

The following statement (in the abstract section) was modified by adding the blue-highlighted phrase: “The properties of fibrous, structural (XRD and mechanical), physical and chemical features were investigated”.

Morphological properties are already present at this study comprising SEL,  fiber length, fiber diameter.

Reviewer’ s Comment 3:

It is also not entirely clear to me how stable the spinning will be due to the presence of impurities in the feedstock.

Author’s reply 3:

Sir, I did not mean anything concerning that stable the spinning will be due to the presence of impurities in the feedstock. In the hardening bath regenerating the fibers, their chemical purification by excluding Cu2+ from fibers. The term “staple” was used to express those short fibers “short fiber–staple”. In textile terminology, "staple fibers" are fibers that are shorter than continuous filament fibers. Common examples of staple fibers include cotton, wool, and certain synthetic fibers.

Cellulose does not have a specific melting point because it is a polymer made up of repeating units of glucose as well as due to its crystalline nature and the presence of extensive hydrogen bonding as well as van der Vaals forces between the cation “H” and the anion “OH”. Instead of melting, cellulose typically decomposes at high temperatures. The decomposition of cellulose typically occurs around 240 to 260°C. Instead of a melting point, it exhibits a transition where it breaks down chemically rather than transitioning from a solid to a liquid state. Yes, that's correct! Cellulose, being a high molecular weight polymer, does not have a conventional melting point like smaller organic compounds. Instead of melting and turning into a liquid, it undergoes thermal degradation when heated at elevated temperatures into smaller molecules, releasing volatile compounds and carbon.

Under an inert atmosphere, the degradation process might occur at slightly higher temperatures, as the lack of oxygen can stabilize the material to some extent. However, cellulose will still not melt; instead, it will continue to degrade chemically, producing smaller volatile compounds, char, and gases. The behavior of cellulose under different thermal conditions can vary based on factors like the degree of polymerization and moisture content, so experimental conditions can lead to different results. Nonetheless, the key point remains that cellulose does not melt but decomposes thermally. It is worth mentioning that the differential thermal analysis (DTA) is a thermal analysis technique that was done under a N2-atmospere in order to measure the temperature and heat flow associated with material transitions as a function of temperature.

For cellulose, DTA does not indicate a melting point in the conventional sense, because cellulose decomposes rather than melting. In DTA experiments, cellulose typically shows an endothermic peak associated with thermal degradation rather than a well-defined melting point. This thermal degradation usually occurs at temperatures between approximately 240 to 350°C), depending on factors such as sample purity, molecular weight, and specific environmental conditions. In some cases, studies on cellulose derivatives or modified cellulose might show some thermoplastic behavior or glass transition behavior at lower temperatures, but for pure cellulose, decomposition is the main concern observed in DTA or other thermal analyses.

Rayon, a regenerated cellulose fiber, does not have a singular melting point like crystalline solids. Instead, it softens and decomposes upon heating. The thermal behavior of rayon is generally characterized by a glass transition temperature (Tg) around 200°C. As the temperature rises, rayon fibers lose strength, and at temperatures above approximately 300°C), rayon tends to decompose rather than melt. In practical applications, rayon behaves similarly to other cellulose derivatives; it will not melt, but it can become pliable or shrink upon heating before eventually breaking down chemically at higher temperatures. Since the Tg of rayon as well as similar cellulose derivatives is important for predicting the fiber's behavior at various temperatures and its performance in textile applications, we preferred expression of thermal transition of rayon by glass transition temperature rather than melting point. The glass transition temperature (Tg) of rayon is generally considered to be around 60–80 °C. However, this Tg can vary depending on the specific formulation and degree of polymerization of the cellulose used to produce the rayon. The glass transition temperature is the temperature at which the amorphous regions of the polymer transition from a brittle, glassy state to a more flexible, rubbery state.

The glass transition temperature (Tg) is a critical point in the thermal behavior of amorphous materials, notably polymers and some glasses. It marks the temperature at which a material transitions from a hard and brittle "glassy" state to a more flexible and rubbery state. Below the Tg, the molecular motion of the polymers is limited, making the material rigid, while above this temperature, increased molecular mobility occurs, allowing for more flexibility.

The Tg is influenced by several factors including:

  1. Composition: Different polymers or glass compositions will have different Tg values.
  2. Molecular Weight: Generally, higher molecular weight polymers have higher Tg due to increased entanglement.
  3. Additives: Plasticizers, fillers, and other additives can significantly affect the Tg; for instance, plasticizers lower the Tg of polymers.
  4. Processing Conditions: The way a material is processed (e.g., cooling rate) can also impact its Tg.

It is measured using techniques like differential scanning calorimetry (DSC) or dynamic mechanical analysis (DMA). Understanding the Tg is crucial for the design and application of materials in fields ranging from packaging to electronics and construction.

DTA is a reliable method for measuring the glass transition temperature of amorphous materials, allowing researchers and engineers to characterize and predict the thermal behavior of materials in various applications.

In addition, results and discussion was modified, see please the revised version.

Table 3 was extended to comprise more properties for rayon.
